# From Observations to States: Latent Time Series Forecasting

Jie Yang [1] [*]   Yifan Hu [2] [*]   Yuante Li [3]   Kexin Zhang [4]   Kaize Ding [4]   Philip S. Yu [1]

## Abstract

Deep learning has achieved strong performance in Time Series Forecasting (TSF). However, we identify a critical representation paradox, termed Latent Chaos: models with accurate predictions often learn latent representations that are temporally disordered and lack continuity. We attribute this to the dominant observation-space forecasting paradigm, where minimizing point-wise errors on noisy and partially observed data encourages shortcut solutions instead of the recovery of underlying system dynamics. To address this, we propose Latent Time Series Forecasting (LatentTSF), a paradigm that shifts TSF from observation regression to latent state prediction. LatentTSF employs an AutoEncoder to project each observation into a learned latent state space and performs forecasting entirely in this space, allowing the model to focus on learning structured temporal dynamics. We provide an information-theoretic analysis showing that the latent objectives can be motivated as surrogates for maximizing mutual information between predicted and ground-truth latent states and future observations. Extensive experiments on widely-used benchmarks confirm that LatentTSF effectively mitigates latent chaos, yielding consistent improvements in both forecasting accuracy and representation quality. Our code is available in https://github.com/Muyiiiii/LatentTSF.

## 1. Introduction

Time Series Forecasting (TSF) is a fundamental problem (Yue et al., 2022; Eldele et al., 2024; Lim & Zohren, 2021) with broad impact in real-world applications such as traffic flow scheduling (Yin et al., 2021; Bai et al., 2020;

*Equal contribution [1]University of Illinois Chicago [2]Tsinghua University [3]Carnegie Mellon University [4]Northwestern University. Correspondence to: Kaize Ding <kaize.ding@northwestern.edu>, Philip S. Yu <psyu@uic.edu>.

*Proceedings of the $43^{rd}$ International Conference on Machine Learning*, Seoul, South Korea. PMLR 306, 2026. Copyright 2026 by the author(s).

Zhang et al., 2024), weather prediction (Liu et al., 2023; Zheng et al., 2015), and financial market analysis (Hu et al., 2025a;b; Yang et al., 2025c; Cheng et al., 2025; Li et al., 2025). In recent years, deep learning has driven remarkable progress in TSF, with representative models spanning a range of backbone architectures, including RNNs (Lin et al., 2025; Zhu et al., 2024), CNNs (Luo & Wang, 2024; Wu et al., 2022; Wang et al., 2023), MLPs (Zeng et al., 2023; Huang et al.; Hu et al., 2025c), and Transformers (Liu et al., 2024; Qiu et al., 2025b; 2026; Ma et al., 2025; 2026a). Despite their architectural diversity, most TSF methods share a common implicit assumption (Kong et al., 2025; Wang et al., 2025d; 2024a): optimizing observation-space forecasting objectives is sufficient for learning the underlying temporal order and dynamical evolution of system states.

However, through a systematic analysis at the representation level, we uncover a paradoxical phenomenon: a TSF model can achieve low forecasting error in the observation space while its internal latent representations lack stable temporal structure. That is, accurate predictions do not necessarily imply that the model has learned coherent temporal evolution, which raises concerns about generalization under noise, as verified in Sec. 5.7. We refer to this phenomenon as **Latent Chaos**, where temporally adjacent embeddings fail to preserve locality, leading to discontinuous latent trajectories and unstable neighborhoods (defined in App. A).

To investigate this phenomenon, we conduct a multi-view analysis on the Electricity dataset (Wu et al., 2021) using iTransformer (Liu et al., 2023) as a representative backbone, as shown in Figure 1 (a-b). ❶ t-SNE (Maaten & Hinton, 2008) visualizations reveal that the original observations exhibit clear temporal locality, where adjacent time steps tend to cluster together. In contrast, latent embeddings learned by the standard iTransformer scatter irregularly across time steps and fail to form continuous trajectories. ❷ Quantitatively, the averaged Euclidean distance between adjacent time steps in the latent space is substantially larger than that of the original observations (94.03 *vs.* 12.94), indicating severe degradation of temporal locality. ❸ Frequency-domain analysis further shows that latent representations distort the spectral structure of the original data, with weakened dominant periodic components. Together, these observations demonstrate that modern TSF models may fail to encode interpretable temporal structure, despite strong

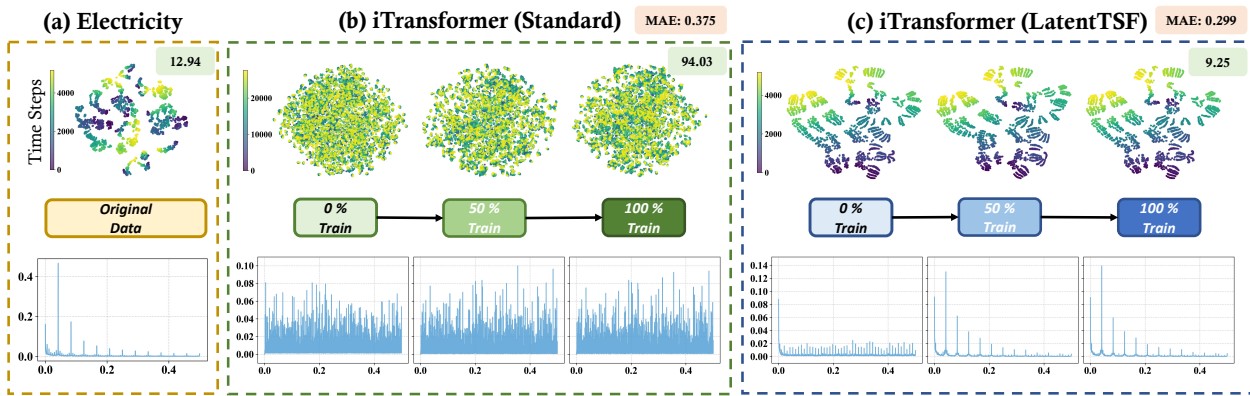

*Figure 1.* **Latent Chaos visualization under LatentTSF.** Electricity dataset: multi-view comparison of **(a)** raw observations, **(b)** standard iTransformer embeddings, and **(c)** iTransformer embeddings trained with LatentTSF, shown at 0%/50%/100% training progress. **Top:** t-SNE visualizations (colored by time index). **Bottom:** frequency-domain spectra of the corresponding representations. **Green box numbers** report the mean normalized Euclidean distance between adjacent time steps, while **Brown box numbers** indicate forecasting MAE. Additional results are provided in App. B.

forecasting accuracy, as reflected by the low MAE values reported in Figure 1 (b).

We argue that Latent Chaos is not incidental, but rather a consequence of the dominant observation-space forecasting paradigm, which can be understood from two complementary perspectives. **(i) System-theoretic perspective:** Real-world observations $\mathbf{X}$ are typically noisy, low-dimensional, and partial projections of underlying high-dimensional dynamical states (Ghugare et al., 2022; Kaiser et al., 2019). Such partial observability implies that crucial latent variables governing system evolution are unavoidably missing or corrupted in the observation space. Consequently, minimizing observation-space prediction errors alone does not guarantee recovery of coherent latent dynamics, and may instead encourage models to fit low-order statistical patterns (e.g., mean reversion, periodic trends, autocorrelation) without recovering the underlying generative dynamics. **(ii) Optimization perspective:** Standard point-wise losses such as mean absolute error (MAE) or mean squared error (MSE) provide limited inductive bias toward temporal continuity (Qiu et al., 2025a; Ma et al., 2026b). These objectives reward accurate numerical predictions but remain agnostic to how latent representations evolve over time. As a result, models tend to prioritize compositional correlations in the input history rather than learning a smooth and identifiable dynamical process (Hu et al., 2026). Together, partial observability at the system level and weak temporal inductive bias at the optimization level reinforce shortcut learning.

These insights raise a central research question: *Is there a training paradigm that leverages partial observations to explicitly guide models toward learning temporally coherent latent dynamics, rather than merely optimizing observation-space accuracy?*

To address this question, we propose **Latent Time Series Forecasting (LatentTSF)**, a general training paradigm that reformulates TSF as latent state prediction rather than direct regression in the observation space. The key idea is to explicitly construct a structured latent state space and learn temporal dynamics within this space, which encourages temporally coherent representations and improves robustness under noisy and partial observations.

Specifically, we first pre-train an AutoEncoder to encode the historical observations at each time step into a sequence of higher-dimensional latent representations, aiming to purify noise and recover missing state information. We then train the standard TSF backbone to predict future latent states from historical ones within the constructed latent space. The decoder is used only to map predicted latent states back to the observation space for final forecasting. Training is guided by a joint prediction and alignment objective defined directly in the latent space. We provide a theoretical analysis showing that the latent objectives can be motivated as surrogates for maximizing the mutual information between predicted representations and ground-truth states and observations, leading to more robust latent dynamics learning. As illustrated in Figure 1 (c), this paradigm effectively restores temporal locality and spectral structure in the learned space, while achieving superior forecasting performance.

Our main contributions can be summarized as follows:

- We identify and quantify **Latent Chaos** in modern TSF models, revealing that despite high accuracy, learned latent representations are temporally disorganized.

- We propose LatentTSF, a general latent-state forecasting paradigm that shifts TSF from observation regression to structured dynamics modeling.

- Extensive experiments confirm consistent gains in prediction accuracy and representation quality across multiple benchmarks, mitigating Latent Chaos.

## 2. Related Work

### 2.1. Deep Learning in Time Series Forecasting

Recent years have witnessed the remarkable success of deep learning in TSF (Hu et al., 2025d; Dai et al., 2024), with a diverse array of backbone architectures proposed to capture complex temporal dependencies. Conventional architectures like RNNs (Lin et al., 2025; Zhu et al., 2024) and CNNs (Luo & Wang, 2024; Wu et al., 2022; Wang et al., 2023) are adapted to capture sequential patterns, while recent advancements have been dominated by Transformer-based models (e.g., PatchTST (Nie, 2022), iTransformer (Liu et al., 2023)), which leverage mechanisms such as patch-based tokenization and attention variants to model long-range dependencies and multivariate correlations. In parallel, MLP-based architectures (e.g., DLinear (Zeng et al., 2023), TimeMixer (Wang et al., 2024b)) have emerged as efficient alternatives, showing that simple linear decomposition can yield robust performance. Despite these architectural innovations, the temporal organization of learned representations remains less explored (Hu et al., 2026; Kim et al., 2021). In particular, most TSF models are primarily optimized for minimizing observation-space forecasting errors, without explicitly enforcing temporal coherence in the latent space (Lu et al., 2025). As a result, latent representations may fail to preserve temporal locality across adjacent time steps, which can limit robustness in complex dynamical settings. This motivates our work LatentTSF towards shifting from observation prediction to latent state prediction, explicitly constructing a structured latent manifold to ensure temporal consistency and improve forecasting performance.

### 2.2. Training Paradigms and Optimization Objectives

Most existing TSF methods follow the standard observation-space forecasting paradigm (Yang et al., 2025a; Tan et al., 2024), where models directly map historical observations to future values by minimizing point-wise losses such as MSE or MAE. While effective, this paradigm can be suboptimal under partial observability (Zhang et al., 2025; Wang et al., 2025b), as real-world observations are often noisy and incomplete projections of underlying system states (Ghugare et al., 2022; Kaiser et al., 2019). To alleviate these issues and improve representation learning, several enhanced training objectives have been explored (Wang et al., 2025c;a; Yu et al., 2024). For example, patchwise or segment-wise losses (Kudrat et al., 2025) introduce structured supervision over local forecasting blocks to better capture temporal patterns beyond point-wise regression. Other approaches incorporate representation-level regularization, such as information-bottleneck-inspired objectives (e.g., Glocal-IB (Yang et al., 2025b)) or alignment-based constraints (e.g., TimeAlign (Hu et al., 2026)) that

leverage target embeddings to shape the latent space, aiming to promote richer semantics and improve training stability. Despite these efforts, most paradigms still optimize forecasting primarily in the observation space or treat latent representations as auxiliary regularizers for the underlying $\mathbf{Y}$ regression, leaving the challenge of explicitly learning temporally coherent latent state dynamics largely underexplored. This further motivates our LatentTSF paradigm, which fundamentally shifts TSF from observation-space regression to latent state prediction, aiming to enable more structured and robust dynamics learning.

## 3. Methodology

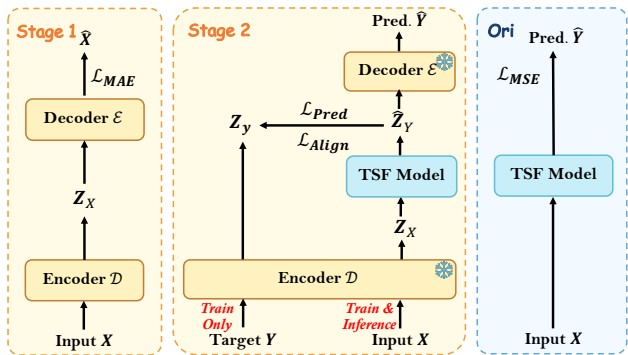

*Figure 2.* **Overview of LatentTSF.** The framework consists of a two-stage pipeline: **(1)** latent state space construction via a pre-trained point-wise AutoEncoder, applied identically to both the lookback window $\mathbf{X}$ and the forecast window $\mathbf{Y}$ to produce $\mathbf{Z}_X = \mathcal{E}(\mathbf{X})$ and $\mathbf{Z}_Y = \mathcal{E}(\mathbf{Y})$ (the figure depicts $\mathbf{X}$ for clarity); and **(2)** latent states forecasting using a TSF backbone, followed by decoding to the observation space.

### 3.1. Problem Definition

#### 3.1.1. STANDARD TIME SERIES FORECASTING

Given a historical multivariate time series $\mathbf{X} = [\mathbf{x}_1, \cdots, \mathbf{x}_L] \in \mathbb{R}^{C \times L}$, where $C$ denotes the number of variables/channels and $L$ is the lookback window length, the goal of TSF is to predict the future sequence $\mathbf{Y} = [\mathbf{x}_{L+1}, \cdots, \mathbf{x}_{L+T}] \in \mathbb{R}^{C \times T}$ over a forecasting horizon $T$. Under the standard TSF paradigm, a model learns a direct mapping $\mathcal{F}_\theta : \mathbb{R}^{C \times L} \to \mathbb{R}^{C \times T}$ by minimizing prediction error in the observation space, typically using point-wise losses such as MAE or MSE.

#### 3.1.2. LATENT TIME SERIES FORECASTING

In practice, observed time series $\mathbf{X}$ are often noisy and partially observable projections of underlying system states, making observation-space objectives prone to shortcut fitting and temporally incoherent representations. To mitigate this issue, we propose LatentTSF, which performs forecasting in a learned latent state space through the

pipeline: $\mathbf{X} \rightarrow \mathbf{Z}_X \rightarrow \widehat{\mathbf{Z}}_Y \rightarrow \widehat{\mathbf{Y}}$.

Specifically, we first construct latent state sequences $\mathbf{Z}_X = \mathcal{E}(\mathbf{X}) = [\mathbf{z}_1, \cdots, \mathbf{z}_L] \in \mathbb{R}^{D \times L}$ and $\mathbf{Z}_Y = \mathcal{E}(\mathbf{Y}) = [\mathbf{z}_{L+1}, \cdots, \mathbf{z}_{L+T}] \in \mathbb{R}^{D \times T}$, using a pre-trained point-wise AutoEncoder with encoder $\mathcal{E}$ and decoder $\mathcal{D}$. Here, $\mathbf{Z}_X$ and $\mathbf{Z}_Y$ denote the latent states encoded from the observations $\mathbf{X}$ and $\mathbf{Y}$, and $D$ is the expanded latent state dimension. Forecasting is then formulated as learning a latent dynamics mapping function $\mathcal{F}_\theta^{\mathbf{Z}} : \mathbb{R}^{D \times L} \rightarrow \mathbb{R}^{D \times T}$, which predicts future latent states $\widehat{\mathbf{Z}}_Y$ by modeling intrinsic temporal dynamics. Finally, the predicted latent states are decoded back to the observation-space forecasts $\widehat{\mathbf{Y}} = \mathcal{D}(\widehat{\mathbf{Z}}_Y)$.

### 3.2. Overview

We propose LatentTSF, a general paradigm that explicitly shifts TSF from observation-space forecasting to latent-space forecasting. As illustrated in Figure 2, LatentTSF consists of two stages: ① **Latent States Space Construction**: We pre-train a point-wise AutoEncoder to map low-dimensional and partial observations to a high-dimensional latent state space, where latent states provide a more expressive and structured space for dynamics modeling. ② **Latent States Forecasting**: A TSF backbone is trained to predict future latent states from historical latent states. Forecasting is performed entirely in the latent space, and the predicted states are subsequently decoded back into the observation space to produce the final forecasts.

### 3.3. Stage 1: Latent State Space Construction

The first challenge is to recover the underlying system states from the observations $\mathbf{X} \in \mathbb{R}^{C \times L}$. We posit that the underlying system states lie on a higher-dimensional manifold where the dynamics are smoother and more regular than in the observation space. To approximate this manifold, we employ a point-wise (rather than temporal) AutoEncoder composed of an encoder $\mathcal{E}$ and a decoder $\mathcal{D}$. For each time step $t$, the encoder projects the observation vector $\mathbf{x}_t \in \mathbb{R}^C$ into an expanded latent state $\mathbf{z}_t \in \mathbb{R}^D$ as follows:

$$\mathbf{z}_t = \mathcal{E}(\mathbf{x}_t), \quad \widehat{\mathbf{x}}_t = \mathcal{D}(\mathbf{z}_t). \tag{1}$$

The latent dimension $D$ may be larger or smaller than $C$ depending on the dataset; the goal is to learn a latent state space that is more conducive to dynamics modeling than the raw observation space, rather than to perform a particular dimensionality change (see App. K). The encoder is pre-trained by minimizing the MAE reconstruction loss for robustness to observation noise:

$$\mathcal{L}_{\text{Rec}} = \frac{1}{L} \sum_{t=1}^{L} \|\mathbf{x}_t - \mathcal{D}(\mathcal{E}(\mathbf{x}_t))\|_1. \tag{2}$$

Once pre-trained, $\mathcal{E}$ and $\mathcal{D}$ are frozen. The encoded sequences $\mathbf{Z}_X = \mathcal{E}(\mathbf{X})$ and $\mathbf{Z}_Y = \mathcal{E}(\mathbf{Y})$ are then used as inputs and targets for latent forecasting.

### 3.4. Stage 2: Latent States Forecasting

In the second stage, we learn the latent transition function $\mathcal{F}_\theta^{\mathbf{Z}}$. Unlike traditional TSF models that learn the mapping $\mathbf{X} \rightarrow \mathbf{Y}$, LatentTSF trains forecasting backbones $\mathbf{Z}_X \rightarrow \mathbf{Z}_Y$ directly in the latent space.

Given historical latent states $\mathbf{Z}_X$, the backbone model $\mathcal{F}_\theta^{\mathbf{Z}}$ (e.g., iTransformer or TimeBase) predicts the future latent states $\widehat{\mathbf{Z}}_Y$ as follows:

$$\widehat{\mathbf{Z}}_Y = \mathcal{F}_\theta^{\mathbf{Z}}(\mathbf{Z}_X) \in \mathbb{R}^{D \times T}. \tag{3}$$

The final forecast $\widehat{\mathbf{Y}}$ is then obtained by decoding the predicted states back to the observation space:

$$\widehat{\mathbf{Y}} = \mathcal{D}(\widehat{\mathbf{Z}}_Y). \tag{4}$$

By constraining the backbone to operate entirely in $\mathbb{R}^D$, we explicitly encourage the model to capture intrinsic system dynamics that are less sensitive to measurement noise and partial observability in $\mathbb{R}^C$.

### 3.5. Training Objective: Latent Alignment & Prediction

A key innovation of LatentTSF is its training objective. Instead of calculating the loss on the final output $\widehat{\mathbf{Y}}$ in observation space, we compute the loss entirely in the latent space. The encoded future sequence $\mathbf{Z}_Y = \mathcal{E}(\mathbf{Y})$ serves as the training target, with the encoder $\mathcal{E}$ fixed. The overall objective combines a prediction loss and an alignment loss:

$$\mathcal{L}_{\text{Total}} = \alpha \cdot \underbrace{\|\mathbf{Z}_Y - \widehat{\mathbf{Z}}_Y\|_F^2}_{\mathcal{L}_{\text{Pred}}} + \beta \cdot \underbrace{\left(1 - \frac{\langle \mathbf{Z}_Y, \widehat{\mathbf{Z}}_Y \rangle_F}{\|\mathbf{Z}_Y\|_F \cdot \|\widehat{\mathbf{Z}}_Y\|_F}\right)}_{\mathcal{L}_{\text{Align}}},$$
$$\tag{5}$$

where $\alpha$ and $\beta$ are balancing hyperparameters, and $\langle \cdot, \cdot \rangle_F$ denotes the Frobenius inner product (equivalent to the dot product of flattened vectors). The prediction term $\mathcal{L}_{\text{Pred}}$ enforces accurate magnitude prediction, while the alignment term $\mathcal{L}_{\text{Align}}$ encourages directional consistency and prevents degenerate scaling solutions.

*Remark* 3.1 (Frozen-encoder anti-collapse). Because $\mathcal{E}$ is frozen, $\mathcal{L}_{\text{Align}}$ regresses $\widehat{\mathbf{Z}}_Y$ toward a stationary target, and a constant prediction $\widehat{\mathbf{z}}_t \equiv \mathbf{c}$ cannot be optimal whenever $\mathcal{E}$ separates distinct inputs, so representation collapse is structurally excluded without negative samples. Unlike SimSiam (Chen & He, 2021) and BYOL (Grill et al., 2020), which rely on stop-gradient or EMA to prevent collapse with learnable targets, our frozen target provides a strictly stronger guarantee (formal argument in App. C.3).

## 4. Theoretical Analysis

As defined in Eq. 5, LatentTSF is trained exclusively with latent-space supervision, using the prediction loss $\mathcal{L}_{\text{Pred}}$ and the alignment loss $\mathcal{L}_{\text{Align}}$. The AutoEncoder is pre-trained and frozen throughout training, so all supervision signals are injected through the latent objectives.

In this section, we provide an information-theoretic justification for this design through the lens of Mutual Information Maximization (MIM) (Voloshynovskiy et al., 2019). We show that the two latent objectives correspond to optimizing tractable surrogates of two complementary mutual information terms (Choi & Lee, 2023): (i) the mutual information between the predicted and ground-truth latent states $I(\mathbf{Z}_Y; \widehat{\mathbf{Z}}_Y)$, and (ii) the mutual information between the predicted states and the future observations, $I(\mathbf{Y}; \widehat{\mathbf{Z}}_Y)$. Formally, the mutual information $I(\mathbf{A}; \mathbf{B})$ of $\mathbf{A}$ and $\mathbf{B}$ (Alemi et al., 2016) is defined as:

$$I(\mathbf{A}; \mathbf{B}) = \mathbb{E}\left[\log \frac{p(\mathbf{A}, \mathbf{B})}{p(\mathbf{A})p(\mathbf{B})}\right]. \tag{6}$$

### 4.1. $\mathcal{L}_{\text{Pred}}$ as an Objective for Maximizing $I(\mathbf{Z}_Y; \widehat{\mathbf{Z}}_Y)$

We first analyze the latent prediction loss $\mathcal{L}_{\text{Pred}}$ (see full derivation in App. C.1). The mutual information between $\mathbf{Z}_Y$ and $\widehat{\mathbf{Z}}_Y$ can be decomposed as follows:

$$\begin{aligned} I(\mathbf{Z}_Y; \widehat{\mathbf{Z}}_Y) &= H(\mathbf{Z}_Y) - H(\mathbf{Z}_Y \mid \widehat{\mathbf{Z}}_Y), \\ &= H(\mathbf{Z}_Y) + \mathbb{E}[\log p(\mathbf{Z}_Y \mid \widehat{\mathbf{Z}}_Y)]. \end{aligned} \tag{7}$$

Since $H(\mathbf{Z}_Y)$ is constant with respect to model parameters, maximizing $I(\mathbf{Z}_Y; \widehat{\mathbf{Z}}_Y)$ reduces to maximizing the conditional log-likelihood $\log p(\mathbf{Z}_Y \mid \widehat{\mathbf{Z}}_Y)$.

As $p(\mathbf{Z}_Y|\widehat{\mathbf{Z}}_Y)$ is generally intractable, we introduce a variational distribution $q_\theta(\mathbf{Z}_Y|\widehat{\mathbf{Z}}_Y)$ (Voloshynovskiy et al., 2019), yielding

$$I(\mathbf{Z}_Y; \widehat{\mathbf{Z}}_Y) \geq \mathbb{E}[\log q_\theta(\mathbf{Z}_Y|\widehat{\mathbf{Z}}_Y)], \tag{8}$$

by the non-negativity of KL divergence. Accordingly, we define $\mathcal{L}_{\text{Pred}}$ as:

$$\mathcal{L}_{\text{Pred}} \overset{\text{def}}{=} -\mathbb{E}[\log q_\theta(\mathbf{Z}_Y \mid \widehat{\mathbf{Z}}_Y)]. \tag{9}$$

We assume the latent prediction error follows an isotropic Gaussian with fixed variance $\sigma^2$ (Kingma & Welling, 2013):

$$\mathbf{Z}_Y = \widehat{\mathbf{Z}}_Y + \epsilon, \quad \text{where } \epsilon \sim \mathcal{N}(\mathbf{0}, \sigma^2 \mathbf{I}). \tag{10}$$

Thus, we have the conditional distribution $q_\theta(\mathbf{Z}_Y \mid \widehat{\mathbf{Z}}_Y) = \mathcal{N}(\widehat{\mathbf{Z}}_Y, \sigma^2 \mathbf{I})$, and $\mathcal{L}_{\text{Pred}}$ can be further reduced as:

$$\mathcal{L}_{\text{Pred}} = \mathbb{E}[\frac{1}{2\sigma^2}\|\mathbf{Z}_Y - \widehat{\mathbf{Z}}_Y\|_F^2 + \frac{D}{2}\log(2\pi\sigma^2)], \tag{11}$$

where $D$ denotes the dimensionality of the latent state $\mathbf{Z}_Y$ and $\|\cdot\|_F$ denotes the Frobenius norm over all elements. Ignoring the constant term $\frac{D}{2}\log(2\pi\sigma^2)$ and the fixed scaling factor $\frac{1}{2\sigma^2}$, maximizing $\mathbb{E}[\log p(\mathbf{Z}_Y \mid \widehat{\mathbf{Z}}_Y)]$ is equivalent to minimizing the squared error objective:

$$\mathcal{L}_{\text{Pred}} \propto \mathbb{E}\left[\|\mathbf{Z}_Y - \widehat{\mathbf{Z}}_Y\|_F^2\right]. \tag{12}$$

In other words, $\mathcal{L}_{\text{Pred}}$ can be viewed as maximizing a conditional log-likelihood term, which encourages $\widehat{\mathbf{Z}}_Y$ to be maximally informative about $\mathbf{Z}_Y$.

### 4.2. $\mathcal{L}_{\text{Align}}$ as an Objective for Maximizing $I(\mathbf{Y}; \widehat{\mathbf{Z}}_Y)$

While one may incorporate observation-level supervision by regressing the decoded output $\widehat{\mathbf{Y}} = \mathcal{D}(\widehat{\mathbf{Z}}_Y)$ toward $\mathbf{Y}$, this strategy often leads to unstable optimization in latent forecasting. Since the decoder is frozen and nonlinear, small latent deviations may be amplified into large reconstruction errors, yielding noisy gradients. Empirically, we observe limited or negative gains from direct observation-space regression in Sec. 5.3.1.

Instead, we introduce an alignment loss $\mathcal{L}_{\text{Align}}$ that maximizes the mutual information between predicted latent states and future observations $I(\mathbf{Y}; \widehat{\mathbf{Z}}_Y)$ (Oord et al., 2018). Starting from the definition (see full derivation in App. C.2):

$$\begin{aligned} I(\mathbf{Y}; \widehat{\mathbf{Z}}_Y) &= \mathbb{E}\left[\log \frac{p(\mathbf{Y} \mid \widehat{\mathbf{Z}}_Y)}{p(\mathbf{Y})}\right], \\ &= \mathbb{E}\left[\log p(\mathbf{Y} \mid \widehat{\mathbf{Z}}_Y) - \log p(\mathbf{Y})\right]. \end{aligned} \tag{13}$$

Directly modeling $\frac{p(\mathbf{Y}|\widehat{\mathbf{Z}}_Y)}{p(\mathbf{Y})}$ is intractable in practice. Inspired by InfoNCE (Oord et al., 2018; He et al., 2020), we define a similarity score function:

$$s_\theta(\mathbf{Y}, \widehat{\mathbf{Z}}_Y) = \frac{\left\langle \widehat{\mathbf{Z}}_Y, \mathcal{E}(\mathbf{Y}) \right\rangle_F}{\|\widehat{\mathbf{Z}}_Y\|_F \cdot \|\mathcal{E}(\mathbf{Y})\|_F}, \tag{14}$$

where $\mathcal{E}$ denotes the frozen encoder of the pre-trained AutoEncoder, and $\langle \cdot, \cdot \rangle_F$ denotes the inner product (vectorized when applicable). We denote $\mathbf{Z}_Y = \mathcal{E}(\mathbf{Y})$.

Using in-batch samples $\mathbf{Y}'$, the InfoNCE objective provides a tractable contrastive estimator and yields a lower bound as follows:

$$\mathcal{L}_{\text{Align}}^{\text{NCE}} = -\mathbb{E}\left[\log \frac{\exp\left(s_\theta(\mathbf{Y}, \widehat{\mathbf{Z}}_Y)\right)}{\sum\limits_{\mathbf{Y}' \in \mathcal{B}} \exp\left(s_\theta(\mathbf{Y}', \widehat{\mathbf{Z}}_Y)\right)}\right], \tag{15}$$

and it satisfies:

$$I(\mathbf{Y}; \widehat{\mathbf{Z}}_Y) \geq \log|\mathcal{B}| - \mathcal{L}_{\text{Align}}^{\text{NCE}}, \tag{16}$$

*Table 1.* **Performance comparison on six benchmark datasets.** We report the average MSE and MAE of six widely used baselines before and after applying LatentTSF. Best results are highlighted. Full results are shown in Tab. 7.

| Models | | CMoS | | | | DLinear | | | | PatchTST | | | | TimeBase | | | | TimeXer | | | | iTransformer | | | |
|---|---|---|---|---|---|---|---|---|---|---|---|---|---|---|---|---|---|---|---|---|---|---|---|---|---|
| | | Original | | w/LatentTSF | | Original | | w/LatentTSF | | Original | | w/LatentTSF | | Original | | w/LatentTSF | | Original | | w/LatentTSF | | Original | | w/LatentTSF | |
| Metric | | MSE | MAE | MSE | MAE | MSE | MAE | MSE | MAE | MSE | MAE | MSE | MAE | MSE | MAE | MSE | MAE | MSE | MAE | MSE | MAE | MSE | MAE | MSE | MAE |
| ETTm1 | *Avg.* | 0.368 | 0.389 | 0.354 | 0.376 | 0.358 | 0.383 | 0.349 | 0.375 | 0.357 | 0.386 | 0.355 | 0.383 | 0.370 | 0.388 | 0.364 | 0.384 | 0.378 | 0.401 | 0.361 | 0.386 | 0.373 | 0.406 | 0.365 | 0.396 |
| ETTm2 | *Avg.* | 0.275 | 0.332 | 0.253 | 0.314 | 0.266 | 0.331 | 0.251 | 0.316 | 0.261 | 0.324 | 0.247 | 0.310 | 0.264 | 0.324 | 0.262 | 0.321 | 0.277 | 0.333 | 0.254 | 0.317 | 0.273 | 0.334 | 0.263 | 0.322 |
| ETTh1 | *Avg.* | 0.433 | 0.441 | 0.421 | 0.426 | 0.428 | 0.445 | 0.411 | 0.428 | 0.437 | 0.456 | 0.409 | 0.435 | 0.420 | 0.423 | 0.410 | 0.416 | 0.485 | 0.489 | 0.432 | 0.454 | 0.485 | 0.486 | 0.441 | 0.459 |
| ETTh2 | *Avg.* | 0.367 | 0.411 | 0.349 | 0.391 | 0.500 | 0.480 | 0.363 | 0.407 | 0.361 | 0.402 | 0.348 | 0.387 | 0.389 | 0.435 | 0.379 | 0.416 | 0.370 | 0.414 | 0.362 | 0.400 | 0.406 | 0.430 | 0.364 | 0.403 |
| Traffic | *Avg.* | 0.559 | 0.365 | 0.510 | 0.342 | 0.525 | 0.373 | 0.552 | 0.376 | 0.982 | 0.629 | 0.719 | 0.486 | 0.591 | 0.396 | 0.539 | 0.356 | 1.270 | 0.721 | 0.636 | 0.438 | 0.798 | 0.526 | 0.672 | 0.464 |
| Electricity | *Avg.* | 0.239 | 0.318 | 0.187 | 0.284 | 0.201 | 0.313 | 0.182 | 0.284 | 0.389 | 0.456 | 0.207 | 0.318 | 0.235 | 0.322 | 0.203 | 0.295 | 0.242 | 0.352 | 0.203 | 0.313 | 0.268 | 0.375 | 0.194 | 0.299 |

where $\mathcal{B}$ denotes the mini-batch. While InfoNCE establishes a rigorous lower bound on mutual information, computing its normalization term over large batches can be computationally expensive. Moreover, since the frozen encoder $\mathcal{E}$ provides a fixed reference manifold, *positive-pair alignment is the dominant driver of feature consistency* in our forecasting setup, while contrastive repulsion of negatives offers diminishing returns (Chen & He, 2021; Grill et al., 2020; Hu et al., 2026). We therefore adopt a simplified objective that maximizes the cosine similarity between predicted and ground-truth representations:

$$\mathcal{L}_{\text{Align}} \stackrel{\text{def}}{=} -\mathbb{E}\left[s_\theta(\mathbf{Y}, \widehat{\mathbf{Z}}_Y)\right]. \tag{17}$$

Discarding the negative-sample normalization formally breaks the InfoNCE bound, so we treat $\mathcal{L}_{\text{Align}}$ as an empirically effective surrogate inspired by mutual information maximization rather than a strict variational bound, with the MI view serving as a design motivation (justified by Remark 3.1) rather than a post-hoc reinterpretation.

As a result, jointly optimizing $\mathcal{L}_{\text{Pred}}$ and $\mathcal{L}_{\text{Align}}$ encourages $\widehat{\mathbf{Z}}_Y$ to be both (i) consistent with the ground-truth latent state $\mathbf{Z}_Y$ and (ii) informative about the future observation $\mathbf{Y}$, leading to improved forecasting accuracy and robustness.

## 5. Experiment

### 5.1. Experiment Setup

**Datasets.** To evaluate the performance and robustness of LatentTSF, we conduct extensive experiments on six widely used real-world benchmarks (Zhou et al., 2021; Wu et al., 2022; Qiu et al., 2024), covering diverse domains including ETTh1, ETTh2, ETTm1, ETTm2, Traffic, and Electricity.

**Baselines.** We verify the effectiveness of LatentTSF by benchmarking it on six representative TSF backbones, including the classical PatchTST (Nie, 2022), iTransformer (Liu et al., 2023), and DLinear (Zeng et al., 2023), as well as the recently proposed TimeXer (Wang et al., 2024c), CMoS (Si et al., 2025), and TimeBase (Huang et al.). In all experiments, we compare the performance of the original observation-space training against our LatentTSF.

**Implementation Details.** We implement our method in PyTorch using a two-stage pipeline, where the AutoEncoder is pre-trained and frozen before optimizing the forecasting backbone in the latent space. More details are in App. D.

### 5.2. Forecasting Performance

As summarized in Tab. 1, LatentTSF achieves strong and consistent performance across diverse benchmarks. Applying our LatentTSF to backbones improves their observation-space counterparts on most datasets in both MSE and MAE. Overall, the results support our central hypothesis: predicting latent states rather than regressing future observations provides a more suitable inductive bias for TSF tasks.

**Impact of Forecasting Horizon.** As shown in Tab. 7, LatentTSF's advantage generally grows with the forecasting horizon, especially at $T = 720$. Under observation-space training, small local errors can compound over long horizons and gradually drift away from plausible dynamics, which we refer to as Latent Chaos. In contrast, LatentTSF forecasts the intrinsic states on a stable latent manifold, implicitly imposing structured temporal regularities. This latent-state modeling mitigates error accumulation and better preserves accuracy and dynamic consistency in long-term forecasting.

**Impact of Variable Dimensionality.** On datasets with many variables, such as Electricity (321) and Traffic (862), LatentTSF yields substantial gains. High-dimensional observations are often noisy, redundant partial projections of the underlying state, making direct regression in the observation space difficult. LatentTSF alleviates this issue by expanding observations into a latent state that is easier to model, and decoding the predicted states to transfer these benefits back

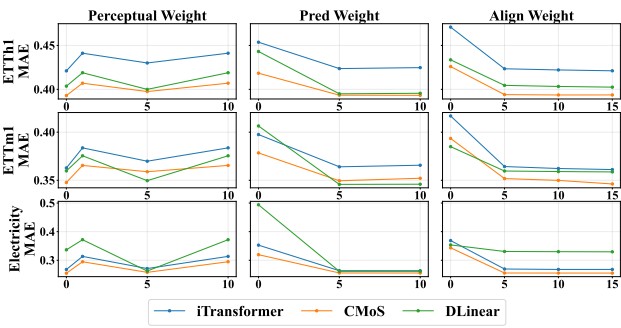

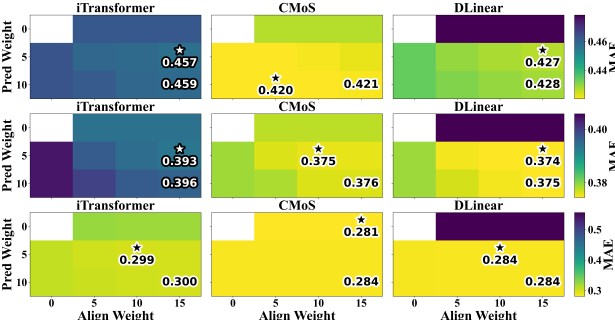

*Figure 3.* **Loss weights sensitivity of LatentTSF.** MAE curves on ETTh1, ETTm1, and Electricity when varying the perceptual, prediction ($\mathcal{L}_{\text{Pred}}$), and alignment ($\mathcal{L}_{\text{Align}}$) weights for three backbones (iTransformer, CMoS, and DLinear).

*Figure 4.* **Interaction between $\mathcal{L}_{\text{Pred}}$ and $\mathcal{L}_{\text{Align}}$.** MAE heatmaps over Pred/Align weight grids on three datasets (rows) and three backbones (columns). Stars mark the best MAE in each subplot.

to the observation space. On lower-dimensional benchmarks such as the ETT family (7), the improvements remain positive but are relatively smaller. This is expected because, in simpler systems, the observation space is closer to the true state space with less distortion, limiting the potential gains from latent reconstruction.

## 5.3. Sensitivity Studies

### 5.3.1. OBSERVATION SPACE: PERCEPTUAL LOSS

To further disentangle the effects of latent-space supervision ($\mathcal{L}_{\text{Pred}}$ and $\mathcal{L}_{\text{Align}}$) from observation-space supervision, we additionally apply an MSE loss on the decoded forecasts $\widehat{\mathbf{Y}}$ (perceptual loss $\mathcal{L}_{\text{Perc}}$). We backpropagate this signal to the forecasting backbone while freezing the AutoEncoder, isolating the impact of the observation-space gradients; a detailed study with trainable encoders/decoders is in Sec. 5.4.

**Sensitivity Analysis.** In Figure 3, we sweep all combinations of loss weights and visualize the averaged results across the remaining settings. Overall, performance of LatentTSF is mainly driven by the latent objectives. Increasing the weight of $\mathcal{L}_{\text{Pred}}$ from 0 to a moderate value (e.g., 5) consistently yields a substantial MAE reduction across datasets and backbones, while further increases to 10 brings only marginal gains, indicating diminishing returns. A similar trend holds for $\mathcal{L}_{\text{Align}}$: raising its weight from 0 to 5 produces a sharp MAE drop, and the curves largely plateau beyond 5–10, suggesting that moderate alignment strength is sufficient to regularize latent dynamics. In contrast, the perceptual loss exhibits non-monotonic and dataset-dependent behavior: small weights can be beneficial in some settings, whereas larger weights may degrade performance.

**Implication.** Consistent with our theoretical analysis that latent-space objectives already provide principled and sufficient supervision for learning predictive state dynamics and prediction targets, we do not enable the perceptual loss by default in the final training recipe of LatentTSF.

### 5.3.2. LATENT SPACE: $\mathcal{L}_{\text{PRED}}$ AND $\mathcal{L}_{\text{ALIGN}}$

**Pred-Align coupling.** Figure 4 visualizes MAE over the Pred–Align weight grid (averaged over other settings), enabling direct inspection of their joint effect. Across datasets and backbones, the best regions typically require both terms: using only one loss generally underperforms, suggesting complementary roles. Concretely, increasing Pred Weight strengthens point-wise latent supervision, while increasing Align Weight further stabilizes latent dynamics via representation-level consistency. Both dimensions exhibit diminishing returns: the optimum lies on a broad plateau rather than a sharp peak, indicating that LatentTSF does not require delicate tuning.

**Backbone- and dataset-dependent sensitivity.** Although the overall trend is consistent, we observe different sensitivity patterns across backbones. For stronger backbones, the MAE landscape is smoother and improvements are more gradual, implying that the backbone already provides a reasonable temporal modeling prior and the latent objectives mainly act as regularizers. In contrast, more linear or lower-capacity models can be more sensitive to the weight coupling: overly weak prediction supervision or misbalanced alignment may lead to suboptimal regions, whereas jointly increasing both weights yields more stable improvements. Across datasets, the relative benefit of stronger alignment is more evident on harder settings, where long-horizon drift or noisy multi-variate structure is more pronounced, aligning with our motivation that latent-space regularization helps prevent trajectory divergence.

**Chosen default weights.** Considering (i) the consistently strong performance of the high-weight plateau across datasets/backbones, and (ii) the need for a simple and unified training paradigm, we set the default hyperparameters to Pred Weight = 10 and Align Weight = 15 for all experiments. This choice lies within the broad optimal region in Figure 4, providing robust performance without per-dataset tuning.

*Table 2.* **AE learning-rate grid: iTransformer on Electricity.** Avg. MSE/MAE over $T \in \{96, 192, 336, 720\}$ under a grid of encoder LR (columns) and AE-decoder LR (rows). **Top:** load a pre-trained AE and fine-tune it. **Bottom:** no pre-trained AE; use 2-layer MLP (AE-style init).

| Model | Dataset | Enc Load | Dec Load | AE Dec LR | Enc LR=0 | | Enc LR=1e-5 | | Enc LR=5e-5 | | Enc LR=1e-4 | | Enc LR=0.001 | |
|---|---|---|---|---|---|---|---|---|---|---|---|---|---|---|
| | | | | | MSE | MAE | MSE | MAE | MSE | MAE | MSE | MAE | MSE | MAE |
| iTransformer | Electricity ( 0.194 / 0.299 ) | ✓ | ✓ | 0 | 0.352 | 0.433 | 0.352 | 0.435 | 0.346 | 0.430 | 0.393 | 0.461 | 0.441 | 0.493 |
| | | | | 1e-5 | 0.306 | 0.406 | 0.306 | 0.405 | 0.297 | 0.397 | 0.457 | 0.506 | 0.423 | 0.485 |
| | | | | 5e-5 | 0.328 | 0.421 | 0.305 | 0.403 | 0.303 | 0.403 | 0.313 | 0.413 | 0.373 | 0.450 |
| | | | | 1e-4 | 0.307 | 0.405 | 0.312 | 0.408 | 0.320 | 0.419 | 0.305 | 0.404 | 0.319 | 0.413 |
| | | | | 0.001 | 0.361 | 0.443 | 0.349 | 0.434 | 0.310 | 0.404 | 0.329 | 0.419 | 0.429 | 0.490 |
| | Electricity ( 0.194 / 0.299 ) | ✗ | ✗ | 0 | 0.372 | 0.450 | 0.365 | 0.445 | 0.364 | 0.444 | 0.385 | 0.459 | 0.452 | 0.507 |
| | | | | 1e-5 | 0.355 | 0.437 | 0.381 | 0.455 | 0.370 | 0.447 | 0.401 | 0.469 | 0.430 | 0.489 |
| | | | | 5e-5 | 0.367 | 0.447 | 0.380 | 0.454 | 0.378 | 0.453 | 0.397 | 0.467 | 0.428 | 0.488 |
| | | | | 1e-4 | 0.388 | 0.461 | 0.351 | 0.434 | 0.392 | 0.465 | 0.372 | 0.448 | 0.443 | 0.501 |
| | | | | 0.001 | 0.351 | 0.432 | 0.387 | 0.457 | 0.364 | 0.441 | 0.398 | 0.464 | 0.438 | 0.496 |

## 5.4. AutoEncoder Analysis

In this section, we analyze the effect of (i) loading a pre-trained AutoEncoder and fine-tuning its encoder/decoder, and (ii) removing pre-training and learning the encoder/decoder-style mappings from scratch. To enable decoder adaptation, we introduce an observation-space MSE loss $\mathcal{L}_{\text{Perc}}$ on the decoded forecasts, and jointly optimize three objectives: $\mathcal{L}_{\text{Perc}}$, $\mathcal{L}_{\text{Pred}}$, and $\mathcal{L}_{\text{Align}}$, with all loss weights set to 10 unless otherwise specified.

**Load Pre-trained AutoEncoder.** We first load the pre-trained AutoEncoder and sweep combinations of encoder/decoder learning rates to fine-tune the latent mapper. Top of Tab. 2 shows that, once the perceptual loss is enabled, the originally stable latent state space becomes perturbed, leading to the same conclusion as in Section 5.3.1. More importantly, the potential benefit from fine-tuning the encoder/decoder cannot compensate for this negative effect: even the best fine-tuned configuration is still noticeably worse than our default setting that uses only $\mathcal{L}_{\text{Pred}}$ and $\mathcal{L}_{\text{Align}}$ with a frozen AutoEncoder. This suggests that decoded-space supervision provides a less reliable training signal and may destabilize the latent manifold.

**Untrained Encoder/Decoder.** We further replace the pre-trained AutoEncoder with two randomly initialized 2-layer MLP modules inserted before and after the forecasting backbone, matching the encoder/decoder-style mapping capacity but without loading any pre-trained parameters. Performance drops even more substantially than the fine-tuning case. We attribute this degradation to the lack of a well-formed latent space: without pre-training, the latent representation is highly unstable and drifts throughout training, which makes the forecasting target non-stationary and yields noisy, inconsistent gradients for the backbone. Consequently, the model fails to learn coherent latent dynamics

and the decoded forecasts deteriorate.

## 5.5. Decoupling Latent Prediction and $\mathcal{L}_{\text{Align}}$

To isolate the contributions of latent-space prediction and the alignment loss, we compare four configurations on DLinear: (a) full LatentTSF, (b) LatentTSF without $\mathcal{L}_{\text{Align}}$, (c) DLinear with $\mathcal{L}_{\text{Align}}$ applied on the observation space, and (d) the vanilla baseline. Tab. 12 shows the ranking $(a) > (b) > (c) \approx (d)$ holds on all five datasets: latent-space prediction is the primary driver, since (b) vs. (d) alone yields an $8.8\%$ MSE reduction on Electricity, while $\mathcal{L}_{\text{Align}}$ provides further refinement only in the latent space and shows negative effects on observations. The two contributions are therefore complementary, not competing.

## 5.6. Latent-Space Dynamical Structure

Beyond visualizations, we quantify the dynamical structure of the latent space along three axes (Tab. 3): CKA between observation and latent representations is far below 1.0 (e.g., 0.015 on ETTh1), ruling out an identity mapping; the Effective Rank of the latent covariance increases substantially (e.g., $7.89 \rightarrow 34.90$ on Electricity, $4.4\times$); and the Temporal Transition Consistency between consecutive latent states improves by $\sim 7\%$ on both ETTh1 ($0.913 \rightarrow 0.983$) and Electricity ($0.894 \rightarrow 0.967$). Together, these confirm that LatentTSF learns a non-trivial, dynamically structured representation rather than a relabeling of the observation space (full results in App. G).

## 5.7. Robustness under Noise and Missing Data

We further evaluate LatentTSF under input perturbations on ETTh1, sweeping additive Gaussian noise with $\sigma \in \{0, 0.1, 0.2, 0.5\}$ and random missing rates in $\{0\%, 10\%, 20\%, 30\%\}$. LatentTSF maintains a lower MSE than the corresponding observation-space training at

*Table 3.* **Quantitative metrics of latent-space dynamical structure.** Lower CKA indicates a non-identity mapping; higher Effective Rank and TTC indicate richer and more coherent latent dynamics. Full results are in App. G.

| Dataset | Space | CKA ↓ | Eff. Rank ↑ | TTC ↑ |
|---|---|---|---|---|
| ETTh1 | Observation | – | 2.86 | 0.913 |
| | **Latent** | 0.015 | **3.36** | **0.983** |
| Electricity | Observation | – | 7.89 | 0.894 |
| | **Latent** | 0.023 | **34.90** | **0.967** |

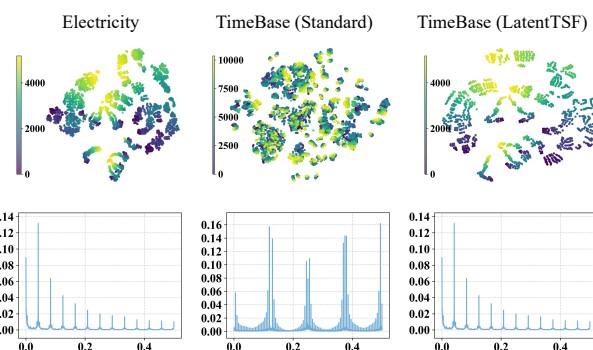

*Figure 5.* **Latent structure of TimeBase on Electricity.** t-SNE (top) and Fourier spectra (bottom) of decoder-pre embeddings from standard training vs. LatentTSF.

every corruption level for both DLinear and PatchTST backbones (full results in App. J). These results provide quantitative support for our claim that LatentTSF encourages representations that go beyond fitting low-order statistical regularities of the clean data: training in a structured latent space yields predictors that degrade more gracefully under realistic data corruption.

### 5.8. Representation Analysis

Figure 1 and Figure 5 compare the representation geometry and frequency characteristics on the Electricity dataset. In the t-SNE plots (top row), the standard observation-space training produces embeddings whose neighboring time steps are heavily interleaved and scattered, showing weak temporal continuity. By contrast, the embeddings produced by LatentTSF exhibit a much clearer time-ordered trajectory: points evolve more smoothly with respect to temporal color, indicating that the learned latent states better preserve sequential structure and reduce the Latent Chaos observed in standard training. The Fourier analysis (bottom row) reveals a consistent pattern: standard training yields a noisier, less coherent spectral profile, whereas LatentTSF produces a more structured and stable spectrum with clearer dominant components and reduced spurious energy. Together, these results suggest that LatentTSF encourages the backbone to learn dynamically consistent latent state evolution, which aligns with our motivation of predicting intrinsic states rather than directly regressing noisy observations. A more comprehensive investigation and additional results regarding the Latent Chaos analysis are provided in App. B.

## 6. Conclusion

In this work, we identify **Latent Chaos** in modern TSF models, where high observation-space accuracy coexists with disordered latent representations lacking temporal coherence. We show that this arises from partial observability and limited temporal inductive bias in standard point-wise loss objectives. To address this, we propose LatentTSF, a paradigm that explicitly constructs a structured latent space and learns temporal dynamics within it, supervised via a joint Alignment and Prediction loss. Theoretically, our latent objectives can be motivated as surrogates for maximizing the mutual information between predicted states and ground-truth states and observations; the frozen target encoder structurally excludes representation collapse, so the non-contrastive form remains principled despite not being a strict variational lower bound. Experimental results illustrate that LatentTSF can restore temporal locality and spectral structure in latent embeddings, improve forecasting accuracy across multiple benchmarks, and degrade more gracefully than observation-space training under input noise and missing data.

## Acknowledgements

This work is supported in part by NSF under grants III-2106758 and POSE-2346158, and by Amazon Research Awards.

## Impact Statement

This paper presents work whose goal is to advance the field of Machine Learning. There are many potential societal consequences of our work, none of which we feel must be specifically highlighted here.

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

# A. Formal Definitions of Key Terms

For clarity, we collect the formal definitions of the three terms used throughout the paper to characterize Latent Chaos.

**Discontinuous trajectories.** Consecutive latent states satisfy $\|\mathbf{z}_{t+1} - \mathbf{z}_t\| \gg \|\mathbf{x}_{t+1} - \mathbf{x}_t\|$ even when the corresponding observations evolve smoothly. That is, small observation-space changes produce disproportionately large latent-space jumps, yielding a non-smooth latent trajectory.

**Unstable neighborhoods.** Similar inputs may map to distant latent points: $\|\mathbf{x} - \mathbf{x}'\| < \epsilon$ does not imply $\|\mathcal{E}(\mathbf{x}) - \mathcal{E}(\mathbf{x}')\|$ is small. Equivalently, the local Lipschitz constant of $\mathcal{E}$ around $\mathbf{x}$ is large, indicating sensitivity rather than robustness in the latent encoding.

**Temporal coherence.** Small (resp. large) observation-space transitions induce small (resp. structured and predictable, not random) latent-space transitions. We do *not* equate temporal coherence with smoothness: genuine regime switches in the underlying dynamics can still produce sharp latent transitions, as long as those transitions remain structured and predictable from the observation-space evidence. This definition allows us to distinguish meaningful sharp events (which a coherent latent space should reflect) from arbitrary discontinuous behavior (which Latent Chaos exhibits).

# B. Latent Chaos Analysis

We extend our multi-view analysis to additional datasets and backbones: iTransformer on ETTh1 and TimeBase on Electricity and ETTh1. Each visualization compares original observations, standard backbone embeddings, and embeddings trained with LatentTSF.

Across all these settings, a consistent pattern emerges. Original observations display smooth, locally coherent trajectories, reflecting the natural temporal structure of the data. In contrast, embeddings learned by the standard backbones scatter irregularly, disrupting temporal continuity and distorting the spectral characteristics of the signals. This instability in the latent space occurs even when the models achieve reasonable forecasting accuracy, highlighting that low prediction errors do not necessarily imply faithful temporal representation. Training with LatentTSF consistently mitigates these effects: latent embeddings become more coherent, preserving both temporal locality and dominant frequency components. Quantitatively, the mean normalized Euclidean distance between consecutive latent states is substantially reduced, and the corresponding MAE improves, demonstrating that the latent dynamics are now both interpretable and predictive. Overall, these results indicate that Latent Chaos is a pervasive phenomenon across backbones and datasets, and that LatentTSF provides a robust mechanism to stabilize latent representations while maintaining effective forecasting performance.

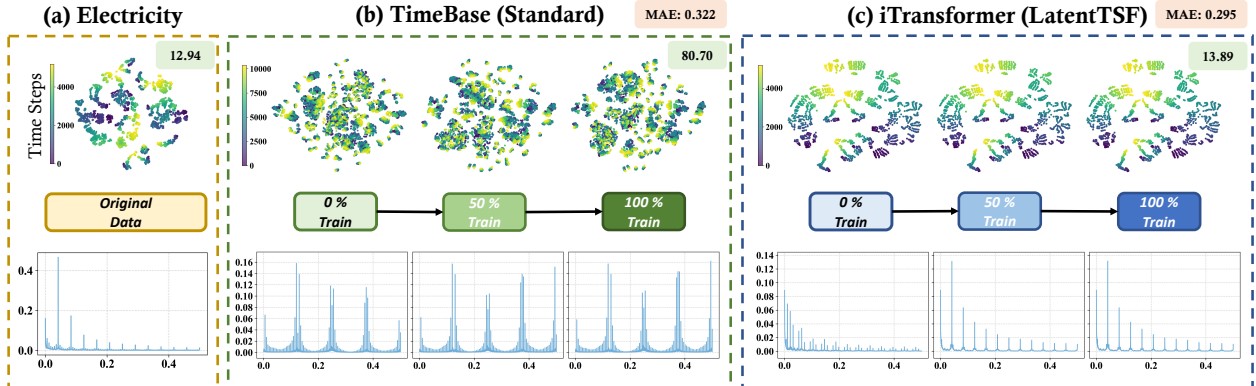

*Figure 6.* **Latent Chaos visualization under LatentTSF.** Electricity dataset with TimeBase: top row shows t-SNE embeddings over training progress (0%/50%/100%), bottom row shows corresponding frequency spectra. Green boxes: mean normalized Euclidean distance between adjacent steps; brown boxes: forecasting MAE.

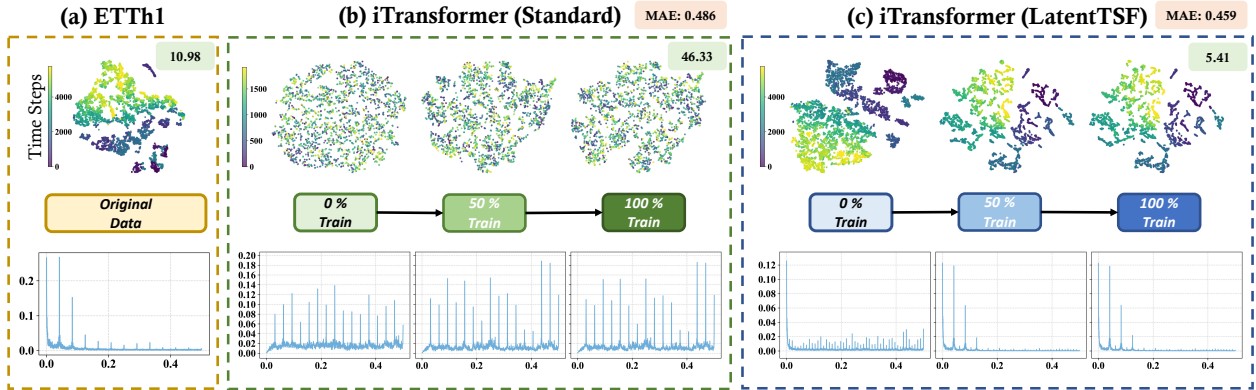

*Figure 7.* **Latent Chaos visualization under LatentTSF.** ETTh1 dataset with iTransformer: top row shows t-SNE embeddings over training progress (0%/50%/100%), bottom row shows corresponding frequency spectra. Green boxes: mean normalized Euclidean distance between adjacent steps; brown boxes: forecasting MAE.

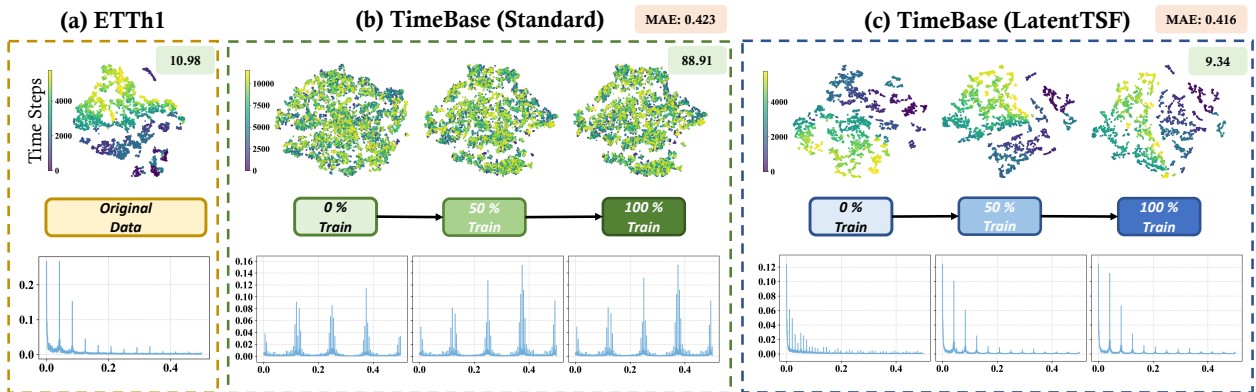

*Figure 8.* **Latent Chaos visualization under LatentTSF.** ETTh1 dataset with TimeBase: top row shows t-SNE embeddings over training progress (0%/50%/100%), bottom row shows corresponding frequency spectra. Green boxes: mean normalized Euclidean distance between adjacent steps; brown boxes: forecasting MAE.

## C. Detailed Theoretical Analysis

### C.1. $\mathcal{L}_{\text{Pred}}$ as an Objective for Maximizing $I(\mathbf{Z}_Y; \widehat{\mathbf{Z}}_Y)$

Following the standard information maximization derivation (Voloshynovskiy et al., 2019), we relate the latent prediction objective to maximizing the mutual information between $\widehat{\mathbf{Z}}_Y$ and $\mathbf{Z}_Y$. Thus, minimizing $\mathcal{L}_{\text{Pred}}$ can be interpreted as maximizing $I(\mathbf{Z}_Y; \widehat{\mathbf{Z}}_Y)$. The mutual information between $\mathbf{Z}_Y$ and $\widehat{\mathbf{Z}}_Y$ can be decomposed as follows:

$$
\begin{aligned}
I(\mathbf{Z}_Y; \widehat{\mathbf{Z}}_Y) &= H(\mathbf{Z}_Y) - H(\mathbf{Z}_Y \mid \widehat{\mathbf{Z}}_Y), \\
&= H(\mathbf{Z}_Y) + \mathbb{E}[\log p(\mathbf{Z}_Y \mid \widehat{\mathbf{Z}}_Y)],
\end{aligned}
\tag{18}
$$

where we use the definition of conditional entropy:

$$
H(\mathbf{Z}_Y | \widehat{\mathbf{Z}}_Y) = -\mathbb{E}[\log p(\mathbf{Z}_Y | \widehat{\mathbf{Z}}_Y)].
\tag{19}
$$

Since $H(\mathbf{Z}_Y)$ is constant with respect to model parameters, maximizing $I(\mathbf{Z}_Y; \widehat{\mathbf{Z}}_Y)$ reduces to maximizing the conditional log-likelihood $\log p(\mathbf{Z}_Y \mid \widehat{\mathbf{Z}}_Y)$. As $p(\mathbf{Z}_Y | \widehat{\mathbf{Z}}_Y)$ is generally intractable, we introduce a variational distribution $q_\theta(\mathbf{Z}_Y | \widehat{\mathbf{Z}}_Y)$, yielding

$$
\begin{aligned}
I(\mathbf{Z}_Y; \widehat{\mathbf{Z}}_Y) &= H(\mathbf{Z}_Y) + \mathbb{E}[\log p(\mathbf{Z}_Y | \widehat{\mathbf{Z}}_Y)], \\
&= H(\mathbf{Z}_Y) + \mathbb{E}[\log p(\mathbf{Z}_Y | \widehat{\mathbf{Z}}_Y)] + \mathbb{E}\left[\log \frac{q_\theta(\mathbf{Z}_Y | \widehat{\mathbf{Z}}_Y)}{q_\theta(\mathbf{Z}_Y | \widehat{\mathbf{Z}}_Y)}\right], \\
&= H(\mathbf{Z}_Y) + \mathbb{E}[\log q_\theta(\mathbf{Z}_Y | \widehat{\mathbf{Z}}_Y)] + \mathbb{E}\left[\log \frac{p(\mathbf{Z}_Y | \widehat{\mathbf{Z}}_Y)}{q_\theta(\mathbf{Z}_Y | \widehat{\mathbf{Z}}_Y)}\right], \\
&\geq \mathbb{E}[\log q_\theta(\mathbf{Z}_Y | \widehat{\mathbf{Z}}_Y)],
\end{aligned}
\tag{20}
$$

by the non-negativity of KL divergence and entropy. Accordingly, we define $\mathcal{L}_{\text{Pred}}$ as:

$$
\mathcal{L}_{\text{Pred}} \stackrel{\text{def}}{=} -\mathbb{E}[\log q_\theta(\mathbf{Z}_Y \mid \widehat{\mathbf{Z}}_Y)].
\tag{21}
$$

We assume the latent prediction error follows an isotropic Gaussian with fixed variance $\sigma^2$ (Kingma & Welling, 2013):

$$
\mathbf{Z}_Y = \widehat{\mathbf{Z}}_Y + \boldsymbol{\epsilon}, \quad \text{where } \boldsymbol{\epsilon} \sim \mathcal{N}(\mathbf{0}, \sigma^2 \mathbf{I}).
\tag{22}
$$

Thus, we have the conditional distribution $q_\theta(\mathbf{Z}_Y \mid \widehat{\mathbf{Z}}_Y) = \mathcal{N}(\widehat{\mathbf{Z}}_Y, \sigma^2 \mathbf{I})$, and $\mathcal{L}_{\text{Pred}}$ can be further reduced as:

$$
\begin{aligned}
\mathcal{L}_{\text{Pred}} &= -\mathbb{E}[\log q_\theta(\mathbf{Z}_Y \mid \widehat{\mathbf{Z}}_Y)], \\
&= \mathbb{E}[\frac{1}{2\sigma^2}\|\mathbf{Z}_Y - \widehat{\mathbf{Z}}_Y\|_F^2 + \frac{D}{2}\log(2\pi\sigma^2)],
\end{aligned}
\tag{23}
$$

where $D$ denotes the dimensionality of the latent state $\mathbf{Z}_Y$ and $\|\cdot\|_F$ denotes the Frobenius norm over all elements. Ignoring the constant term $\frac{D}{2}\log(2\pi\sigma^2)$ and the fixed scaling factor $\frac{1}{2\sigma^2}$, maximizing $\mathbb{E}[\log p(\mathbf{Z}_Y \mid \widehat{\mathbf{Z}}_Y)]$ is equivalent to minimizing the squared error objective:

$$
\mathcal{L}_{\text{Pred}} \propto \mathbb{E}\left[\|\mathbf{Z}_Y - \widehat{\mathbf{Z}}_Y\|_F^2\right].
\tag{24}
$$

In other words, $\mathcal{L}_{\text{Pred}}$ can be viewed as maximizing a conditional log-likelihood term, which encourages $\widehat{\mathbf{Z}}_Y$ to be maximally informative about $\mathbf{Z}_Y$.

### C.2. $\mathcal{L}_{\text{Align}}$ as an Objective for Maximizing $I(\mathbf{Y}; \widehat{\mathbf{Z}}_Y)$.

We introduce an alignment loss $\mathcal{L}_{\text{Align}}$ that maximizes the mutual information between predicted latent states and future observations $I(\mathbf{Y}; \widehat{\mathbf{Z}}_Y)$ (Oord et al., 2018). Starting from the definition:

$$
\begin{aligned}
I(\mathbf{Y}; \widehat{\mathbf{Z}}_Y) &= \mathbb{E}\left[\log \frac{p(\mathbf{Y}, \widehat{\mathbf{Z}}_Y)}{p(\mathbf{Y})p(\widehat{\mathbf{Z}}_Y)}\right], \\
&= \mathbb{E}\left[\log \frac{p(\mathbf{Y} \mid \widehat{\mathbf{Z}}_Y)}{p(\mathbf{Y})}\right].
\end{aligned}
\tag{25}
$$

Directly modeling $\frac{p(\mathbf{Y}|\widehat{\mathbf{Z}}_Y)}{p(\mathbf{Y})}$ is intractable in practice. Inspired by InfoNCE (Oord et al., 2018; He et al., 2020), we define a similarity score function:

$$s_\theta(\mathbf{Y}, \widehat{\mathbf{Z}}_Y) = \frac{\left\langle \widehat{\mathbf{Z}}_Y, \mathcal{E}(\mathbf{Y}) \right\rangle}{\|\widehat{\mathbf{Z}}_Y\|_F \cdot \|\mathcal{E}(\mathbf{Y})\|_F}, \tag{26}$$

where $\mathcal{E}$ denotes the frozen encoder of the pre-trained AutoEncoder, and $\langle \cdot, \cdot \rangle$ denotes the inner product (vectorized when applicable). We denote $\mathbf{Z}_Y = \mathcal{E}(\mathbf{Y})$.

$$
\begin{aligned}
I(\mathbf{Y}; \widehat{\mathbf{Z}}_Y) &= \mathbb{E}\left[\log \frac{p(\mathbf{Y} \mid \widehat{\mathbf{Z}}_Y)}{p(\mathbf{Y})}\right] \\
&= -\mathbb{E}\left[\log \frac{p(\mathbf{Y})}{p(\mathbf{Y} \mid \widehat{\mathbf{Z}}_Y)}\right] \\
&= -\mathbb{E}\left[\log \left(\frac{p(\mathbf{Y})}{p(\mathbf{Y} \mid \widehat{\mathbf{Z}}_Y)} \cdot K\right) - \log K\right] \\
&\approx -\mathbb{E}\left[\log \left(\frac{p(\mathbf{Y})}{p(\mathbf{Y} \mid \widehat{\mathbf{Z}}_Y)} \cdot K\right)\right] \\
&\geq -\mathbb{E}\left[\log \left(1 + \frac{p(\mathbf{Y})}{p(\mathbf{Y} \mid \widehat{\mathbf{Z}}_Y)} \cdot (K-1) \cdot 1\right)\right] \\
&= -\mathbb{E}\left[\log \left(1 + \frac{p(\mathbf{Y})}{p(\mathbf{Y} \mid \widehat{\mathbf{Z}}_Y)} \cdot (K-1) \cdot \mathbb{E}_{p(\mathbf{Y}^-)}\left(\frac{p(\mathbf{Y}^- \mid \widehat{\mathbf{Z}}_Y)}{p(\mathbf{Y}^-)}\right)\right)\right] \\
&= \mathbb{E}\left[\log \frac{\frac{p(\mathbf{Y}|\widehat{\mathbf{Z}}_Y)}{p(\mathbf{Y})}}{\frac{p(\mathbf{Y}|\widehat{\mathbf{Z}}_Y)}{p(\mathbf{Y})} + \sum_{\mathbf{Y}^- \in \mathcal{B} \backslash \{\mathbf{Y}^+\}} \frac{p(\mathbf{Y}^-|\widehat{\mathbf{Z}}_Y)}{p(\mathbf{Y}^-)}}\right] \\
&= \mathbb{E}\left[\log \frac{s_\theta(\mathbf{Y}, \widehat{\mathbf{Z}}_Y)}{s_\theta(\mathbf{Y}, \widehat{\mathbf{Z}}_Y) + \sum_{\mathbf{Y}^- \in \mathcal{B} \backslash \{\mathbf{Y}\}} s_\theta(\mathbf{Y}^-, \widehat{\mathbf{Z}}_Y)}\right],
\end{aligned}
\tag{27}
$$

where $\mathbf{Y}^+$ and $\mathbf{Y}^-$ are positive and negative sample of the mini-batch. Using in-batch samples $\mathbf{Y}'$, the InfoNCE objective provides a tractable contrastive estimator and yields a lower bound as follows:

$$\mathcal{L}_{\text{Align}}^{\text{NCE}} = -\mathbb{E}\left[\log \frac{\exp\left(s_\theta(\mathbf{Y}, \widehat{\mathbf{Z}}_Y)\right)}{\sum_{\mathbf{Y}' \in \mathcal{B}} \exp\left(s_\theta(\mathbf{Y}', \widehat{\mathbf{Z}}_Y)\right)}\right], \tag{28}$$

and it satisfies:

$$I(\mathbf{Y}; \widehat{\mathbf{Z}}_Y) \geq \log |\mathcal{B}| - \mathcal{L}_{\text{Align}}^{\text{NCE}}, \tag{29}$$

where $\mathcal{B}$ denotes the mini-batch. While InfoNCE establishes a rigorous lower bound on mutual information, computing the normalization term over large batches can be computationally expensive. Inspired by non-contrastive methods like SimSiam (Chen & He, 2021) and BYOL (Grill et al., 2020), we hypothesize that the alignment of positive pairs is the primary driver for feature consistency in our specific forecasting setup (Hu et al., 2026). Thus, we adopt a simplified objective that maximizes the cosine similarity between predicted and ground-truth representations:

$$\mathcal{L}_{\text{Align}} \stackrel{\text{def}}{=} -\mathbb{E}\left[s_\theta(\mathbf{Y}, \widehat{\mathbf{Z}}_Y)\right]. \tag{30}$$

Minimizing $\mathcal{L}_{\text{Align}}$ therefore acts as a tractable surrogate for maximizing $I(\mathbf{Y}; \widehat{\mathbf{Z}}_Y)$, encouraging predicted latent states to preserve observation-level information.

As a result, jointly optimizing $\mathcal{L}_{\text{Pred}}$ and $\mathcal{L}_{\text{Align}}$ encourages $\widehat{\mathbf{Z}}_Y$ to be both (i) consistent with the ground-truth latent state $\mathbf{Z}_Y$ and (ii) informative about the future observation $\mathbf{Y}$, leading to improved forecasting accuracy and robustness.

### C.3. Formal Argument for Remark 3.1: Frozen-Encoder Anti-Collapse

We now formalize the claim from Remark 3.1 that representation collapse is structurally excluded under our frozen-encoder design. Let $\mathcal{E} : \mathcal{X} \to \mathbb{R}^D$ denote the pre-trained, frozen encoder, and recall the cosine alignment loss used by our method:

$$\mathcal{L}_{\text{Align}} = - \mathbb{E}_{\mathbf{x}_t \sim p(\mathbf{x})} \left[ \frac{\widehat{\mathbf{z}}_t^\top \mathcal{E}(\mathbf{x}_t)}{\|\widehat{\mathbf{z}}_t\|_2 \, \|\mathcal{E}(\mathbf{x}_t)\|_2} \right]. \tag{31}$$

Suppose, for contradiction, that the predictor converges to a degenerate constant solution $\widehat{\mathbf{z}}_t \equiv \mathbf{c}$ for some non-zero $\mathbf{c} \in \mathbb{R}^D$ and all inputs $\mathbf{x}_t$. Then

$$\mathcal{L}_{\text{Align}}\big|_{\widehat{\mathbf{z}}_t \equiv \mathbf{c}} = - \mathbb{E} \left[ \frac{\mathbf{c}^\top \mathcal{E}(\mathbf{x}_t)}{\|\mathbf{c}\|_2 \, \|\mathcal{E}(\mathbf{x}_t)\|_2} \right]. \tag{32}$$

Since $\cos(\mathbf{c}, \mathcal{E}(\mathbf{x}_t)) \leq 1$ pointwise, with equality iff $\mathcal{E}(\mathbf{x}_t) = \alpha_t \mathbf{c}$ for some $\alpha_t > 0$, the constant solution achieves $\mathcal{L}_{\text{Align}}\big|_{\widehat{\mathbf{z}}_t \equiv \mathbf{c}} = -1$ *only if* $\mathcal{E}(\mathbf{x}_t)$ lies on a single ray through the origin for $p(\mathbf{x})$-almost every $\mathbf{x}_t$, i.e., the encoder maps the input distribution into a one-dimensional subspace up to a $p(\mathbf{x})$-null set. This contradicts the standard requirement on a useful pre-trained AutoEncoder, whose decoder must reconstruct distinct observations and therefore relies on a latent representation of effective rank greater than one (verified empirically in App. G). Hence $\mathcal{L}_{\text{Align}}\big|_{\widehat{\mathbf{z}}_t \equiv \mathbf{c}} > -1$ for any constant $\mathbf{c}$. By contrast, the non-degenerate predictor $\widehat{\mathbf{z}}_t = \mathcal{E}(\mathbf{x}_t)$ attains $\mathcal{L}_{\text{Align}} = -1$ exactly. Therefore no constant solution can be a minimizer, and representation collapse is structurally excluded.

This contrasts with SimSiam (Chen & He, 2021) and BYOL (Grill et al., 2020), where both the online and target networks are learnable: the analogous argument fails because the target $\mathcal{E}(\mathbf{x}_t)$ itself can drift toward a one-dimensional subspace during training, and stop-gradient or EMA approximations are required to break this symmetry. In our setting the target is fixed by pre-training, so the non-contrastive cosine form retains a well-defined optimization target without any architectural mitigation.

## D. Experiment Settings

### D.1. Datasets

We conduct extensive experiments on six widely-used time series datasets and report the statistics in Tab. 4. Detailed descriptions of these datasets are as follows:

(1) **ETT** (Electricity Transformer Temperature) dataset (Zhou et al., 2021) encompasses temperature and power load data from electricity transformers in two regions of China, spanning from 2016 to 2018. This dataset has two granularity levels: ETTh (hourly) and ETTm (15 minutes).

(2) **Electricity** dataset (Wu et al., 2022) features hourly electricity consumption records in kilowatt-hours (kWh) for 321 clients. Sourced from the UCL Machine Learning Repository, this dataset covers the period from 2012 to 2014, providing valuable insights into consumer electricity usage patterns.

(3) **Traffic** dataset (Wu et al., 2022) includes data on hourly road occupancy rates, gathered by 862 detectors across the freeways of the San Francisco Bay area. This dataset, covering the years 2015 to 2016, offers a detailed snapshot of traffic flow and congestion.

### D.2. Baselines

To evaluate the performance of our proposed method, we compare it against a diverse set of state-of-the-art baselines. These models are categorized into two groups: **Classical TSF backbones**, which represent the foundational Transformer-based and linear-based architectures in long-term series forecasting, and **Latest TSF backbones**, which incorporate recent innovations such as exogenous variable integration, spatial-temporal chunking, and ultra-lightweight basis extraction.

⇛ **Classical TSF backbone.**

*Table 4.* **Dataset detailed descriptions.** "Dataset Size" denotes the total number of time points in (Train, Validation, Test) split, respectively. "Prediction Length" denotes the future time points to be predicted. "Frequency" denotes the sampling interval of time points.

| Dataset | Dim | Prediction Length | Dataset Size | Frequency |
|---------|-----|-------------------|--------------|-----------|
| ETTm1 | 7 | $\{96, 192, 336, 720\}$ | $(34465, 11521, 11521)$ | 15 min |
| ETTm2 | 7 | $\{96, 192, 336, 720\}$ | $(34465, 11521, 11521)$ | 15 min |
| ETTh1 | 7 | $\{96, 192, 336, 720\}$ | $(8545, 2881, 2881)$ | 1 hour |
| ETTh2 | 7 | $\{96, 192, 336, 720\}$ | $(8545, 2881, 2881)$ | 1 hour |
| Electricity | 321 | $\{96, 192, 336, 720\}$ | $(18317, 2633, 5261)$ | 1 hour |
| Traffic | 862 | $\{96, 192, 336, 720\}$ | $(12185, 1757, 3509)$ | 1 hour |
| Weather | 21 | $\{96, 192, 336, 720\}$ | $(36792, 5271, 10540)$ | 10 min |

- **PatchTST** (Nie, 2022): A Transformer-based time series model that uses patching and channel independence techniques. It supports effective pretraining and transfer learning across datasets. Source code is available at: `https://github.com/yuqinie98/PatchTST`.

- **iTransformer** (Liu et al., 2023): A Transformer-based time series model that embeds each time series as variable tokens, improving parameter efficiency and modeling precision. It is suitable for long-sequence modeling tasks. Source code is available at: `https://github.com/thuml/iTransformer`.

- **DLinear** (Zeng et al., 2023): A simple yet effective MLP-based model that decomposes a time series into trend and remainder components using a moving average kernel. Each component is then modeled by a single linear layer, challenging the dominance of complex Transformers in long-term forecasting tasks. Source code is available at: `https://github.com/cure-lab/LTSF-Linear`.

⇥ **Latest TSF backbone.**

- **TimeXer** (Wang et al., 2024c): A Transformer-based model specifically designed to empower time series forecasting by effectively incorporating exogenous variables. It employs a variable-specific embedding strategy and a cross-variable attention mechanism to capture the complex correlations between target series and various external factors, while maintaining linear complexity. Source code is available at: `https://github.com/thuml/TimeXer`.

- **CMoS** (Si et al., 2025): A novel time series prediction framework that rethinks forecasting through the lens of chunk-wise spatial correlations. It divides time series into chunks to capture both intra-chunk temporal patterns and inter-variable spatial dependencies simultaneously, addressing the limitations of point-wise or global correlation modeling. Source code is available at: `https://github.com/CSTCloudOps/CMoS`.

- **TimeBase** (Huang et al.): An ultra-lightweight forecasting framework that leverages the low-rank structure of time series. It achieves extreme efficiency by extracting core basis temporal components and transforming traditional point-level forecasting into segment-level prediction. Source code is available at: `https://github.com/hqh0728/TimeBase`.

### D.3. Metric Details

Regarding metrics, we utilize the mean square error (MSE) and mean absolute error (MAE) for long-term forecasting. The calculations of these metrics are:

$$MSE = \frac{1}{T} \sum_{0}^{T} (\hat{y}_i - y_i)^2, \qquad MAE = \frac{1}{T} \sum_{0}^{T} |\hat{y}_i - y_i|$$

## D.4. Hyperparameter Settings

All experiments are conducted on one NVIDIA 4090 24GB GPU. We use the Adam optimizer with a learning rate selected from $\{1e^{-2}, 1e^{-3}\}$. The batch size is set to 32 for the Electricity and Traffic datasets, and 256 for all other datasets. Tab. 5 provides detailed settings for our two-stage paradigm and Tab. 6 provides detailed hyperparameter settings for each dataset.

*Table 5.* **Common hyperparameters.** This table summarizes the global settings used across all datasets, including AutoEncoder specifications and loss balancing terms.

| Parameter | Value | Parameter | Value |
|---|---|---|---|
| *Training & Backbone* | | | |
| Seq Len ($L$) | 720 | Layers (E/D) | 2 / 1 |
| Pred Lens ($T$) | $\{96, 192, 336, 720\}$ | N-Heads | 4 |
| Train Epochs | 100 | Dropout | 0.1 |
| Patience | 5 | LR Scheduler | Cosine |
| *AutoEncoder & Objectives* | | | |
| AE Type | MLP | MSE Weight ($\alpha$) | 10.0 |
| AE Seq Len | 24 | Align Weight ($\beta$) | 15.0 |
| AE Loss | MAE | Freeze AE | True |

*Table 6.* **Dataset-specific hyperparameters.** Detailed architecture dimensions and training settings for each dataset.

| Dataset | Model Architecture | | | Training Optimization | |
|---|---|---|---|---|---|
| | Enc In | $d_{\text{model}}$ | $d_{\text{ff}}$ | Batch Size | Learning Rate |
| ETTh1 | 7 | 32 | 64 | 256 | 0.0003 |
| ETTh2 | 7 | 64 | 128 | 256 | 0.0003 |
| ETTm1 | 7 | 32 | 64 | 256 | 0.0003 |
| ETTm2 | 7 | 64 | 128 | 256 | 0.0003 |
| Electricity | 321 | 512 | 1024 | 32 | 0.0010 |
| Traffic | 862 | 512 | 1024 | 32 | 0.0010 |

## E. Full Results

Tab. 7 shows the impact of applying LatentTSF across six benchmark datasets and multiple forecasting horizons. In terms of effectiveness, applying LatentTSF generally enhances forecasting accuracy by modeling latent states rather than raw observations, reducing error accumulation and improving long-term prediction stability. For instance, on Electricity, DLinear achieves the strongest baseline performance, and LatentTSF further reduces errors, demonstrating that even high-performing models can benefit from latent-state training. On ETTh1, iTransformer exhibits notable gains at longer horizons, highlighting how LatentTSF can stabilize predictions and mitigate error accumulation. In addition, the compatibility of LatentTSF with existing forecasting backbones arises from its ability to seamlessly integrate with both linear and transformer-style models. This design offers a practical framework that leverages the strengths of the underlying architecture, allowing it to improve performance without requiring architectural modifications. Such compatibility ensures that LatentTSF remains flexible and adaptable across datasets with varying dimensionality and temporal complexity.

## F. Additional AutoEncoder Analysis

Across additional backbones and datasets, including iTransformer on ETTh1 and DLinear on both Electricity and ETTh1 (Tab. 8, 9, and 10), we observe trends fully consistent with the main results reported in Section 5.4. When loading a pre-trained AutoEncoder, introducing decoded-space supervision and fine-tuning the encoder/decoder generally degrades forecasting performance. Removing pre-training and learning encoder/decoder-style mappings from scratch leads to even more pronounced performance drops across all settings, indicating severe instability in the learned latent representations.

*Table 7.* **Performance comparison on six benchmark datasets.** We report the full MSE and MAE of six widely used baselines before and after applying LatentTSF. Best results are highlighted .

| Models | Metric | CMoS Original MSE | CMoS Original MAE | CMoS w/LatentTSF MSE | CMoS w/LatentTSF MAE | DLinear Original MSE | DLinear Original MAE | DLinear w/LatentTSF MSE | DLinear w/LatentTSF MAE | PatchTST Original MSE | PatchTST Original MAE | PatchTST w/LatentTSF MSE | PatchTST w/LatentTSF MAE | TimeBase Original MSE | TimeBase Original MAE | TimeBase w/LatentTSF MSE | TimeBase w/LatentTSF MAE | TimeXer Original MSE | TimeXer Original MAE | TimeXer w/LatentTSF MSE | TimeXer w/LatentTSF MAE | iTransformer Original MSE | iTransformer Original MAE | iTransformer w/LatentTSF MSE | iTransformer w/LatentTSF MAE |
|---|---|---|---|---|---|---|---|---|---|---|---|---|---|---|---|---|---|---|---|---|---|---|---|---|---|
| ETTm1 96 | | 0.316 | 0.362 | 0.302 | 0.346 | 0.307 | 0.351 | 0.298 | 0.344 | 0.295 | 0.350 | 0.294 | 0.348 | 0.336 | 0.370 | 0.327 | 0.363 | 0.323 | 0.370 | 0.313 | 0.358 | 0.309 | 0.367 | 0.313 | 0.361 |
| ETTm1 192 | | 0.344 | 0.375 | 0.334 | 0.364 | 0.339 | 0.372 | 0.329 | 0.364 | 0.345 | 0.378 | 0.344 | 0.374 | 0.351 | 0.378 | 0.345 | 0.373 | 0.356 | 0.387 | 0.338 | 0.374 | 0.346 | 0.390 | 0.341 | 0.382 |
| ETTm1 336 | | 0.381 | 0.396 | 0.364 | 0.382 | 0.367 | 0.388 | 0.359 | 0.381 | 0.370 | 0.394 | 0.367 | 0.390 | 0.374 | 0.391 | 0.369 | 0.387 | 0.384 | 0.404 | 0.368 | 0.391 | 0.385 | 0.414 | 0.374 | 0.403 |
| ETTm1 720 | | 0.429 | 0.421 | 0.417 | 0.412 | 0.417 | 0.420 | 0.408 | 0.410 | 0.419 | 0.426 | 0.416 | 0.421 | 0.419 | 0.415 | 0.415 | 0.412 | 0.450 | 0.441 | 0.425 | 0.422 | 0.453 | 0.453 | 0.433 | 0.438 |
| ETTm1 *Avg.* | | 0.368 | 0.389 | 0.354 | 0.376 | 0.358 | 0.383 | 0.349 | 0.375 | 0.357 | 0.386 | 0.355 | 0.383 | 0.370 | 0.388 | 0.364 | 0.384 | 0.378 | 0.401 | 0.361 | 0.386 | 0.373 | 0.406 | 0.365 | 0.396 |
| ETTm2 96 | | 0.181 | 0.270 | 0.164 | 0.254 | 0.163 | 0.256 | 0.161 | 0.254 | 0.168 | 0.258 | 0.158 | 0.248 | 0.184 | 0.272 | 0.181 | 0.268 | 0.186 | 0.268 | 0.168 | 0.259 | 0.183 | 0.278 | 0.173 | 0.263 |
| ETTm2 192 | | 0.249 | 0.314 | 0.217 | 0.291 | 0.217 | 0.295 | 0.215 | 0.290 | 0.242 | 0.307 | 0.214 | 0.288 | 0.233 | 0.305 | 0.230 | 0.301 | 0.244 | 0.312 | 0.221 | 0.295 | 0.240 | 0.315 | 0.231 | 0.302 |
| ETTm2 336 | | 0.296 | 0.345 | 0.273 | 0.329 | 0.293 | 0.353 | 0.268 | 0.328 | 0.281 | 0.341 | 0.265 | 0.323 | 0.280 | 0.336 | 0.278 | 0.331 | 0.296 | 0.348 | 0.275 | 0.331 | 0.295 | 0.350 | 0.284 | 0.336 |
| ETTm2 720 | | 0.375 | 0.399 | 0.356 | 0.382 | 0.392 | 0.420 | 0.362 | 0.394 | 0.352 | 0.389 | 0.351 | 0.382 | 0.357 | 0.385 | 0.358 | 0.382 | 0.380 | 0.403 | 0.354 | 0.383 | 0.373 | 0.396 | 0.365 | 0.386 |
| ETTm2 *Avg.* | | 0.275 | 0.332 | 0.253 | 0.314 | 0.266 | 0.331 | 0.251 | 0.316 | 0.261 | 0.324 | 0.247 | 0.310 | 0.264 | 0.324 | 0.262 | 0.321 | 0.277 | 0.333 | 0.254 | 0.317 | 0.273 | 0.334 | 0.263 | 0.322 |
| ETTh1 96 | | 0.412 | 0.429 | 0.400 | 0.413 | 0.375 | 0.402 | 0.366 | 0.392 | 0.372 | 0.408 | 0.370 | 0.404 | 0.367 | 0.387 | 0.363 | 0.384 | 0.417 | 0.444 | 0.391 | 0.421 | 0.397 | 0.429 | 0.392 | 0.422 |
| ETTh1 192 | | 0.421 | 0.428 | 0.419 | 0.415 | 0.412 | 0.425 | 0.405 | 0.416 | 0.435 | 0.450 | 0.403 | 0.425 | 0.414 | 0.413 | 0.410 | 0.410 | 0.444 | 0.462 | 0.418 | 0.439 | 0.438 | 0.456 | 0.420 | 0.441 |
| ETTh1 336 | | 0.437 | 0.436 | 0.438 | 0.428 | 0.440 | 0.450 | 0.435 | 0.437 | 0.456 | 0.466 | 0.421 | 0.441 | 0.447 | 0.431 | 0.439 | 0.425 | 0.507 | 0.501 | 0.451 | 0.463 | 0.503 | 0.493 | 0.450 | 0.463 |
| ETTh1 720 | | 0.463 | 0.471 | 0.428 | 0.449 | 0.483 | 0.502 | 0.436 | 0.468 | 0.483 | 0.498 | 0.441 | 0.471 | 0.451 | 0.460 | 0.429 | 0.446 | 0.572 | 0.548 | 0.470 | 0.493 | 0.602 | 0.566 | 0.501 | 0.510 |
| ETTh1 *Avg.* | | 0.433 | 0.441 | 0.421 | 0.426 | 0.428 | 0.445 | 0.411 | 0.428 | 0.437 | 0.456 | 0.409 | 0.435 | 0.420 | 0.423 | 0.410 | 0.416 | 0.485 | 0.489 | 0.432 | 0.454 | 0.485 | 0.486 | 0.441 | 0.459 |
| ETTh2 96 | | 0.319 | 0.371 | 0.293 | 0.348 | 0.294 | 0.361 | 0.275 | 0.342 | 0.285 | 0.348 | 0.273 | 0.334 | 0.367 | 0.416 | 0.353 | 0.399 | 0.296 | 0.364 | 0.289 | 0.348 | 0.310 | 0.366 | 0.293 | 0.351 |
| ETTh2 192 | | 0.375 | 0.413 | 0.352 | 0.386 | 0.408 | 0.435 | 0.349 | 0.390 | 0.360 | 0.394 | 0.345 | 0.378 | 0.360 | 0.405 | 0.356 | 0.391 | 0.352 | 0.396 | 0.365 | 0.396 | 0.407 | 0.423 | 0.368 | 0.398 |
| ETTh2 336 | | 0.367 | 0.413 | 0.364 | 0.403 | 0.496 | 0.490 | 0.411 | 0.439 | 0.385 | 0.419 | 0.367 | 0.398 | 0.428 | 0.470 | 0.416 | 0.445 | 0.381 | 0.424 | 0.388 | 0.417 | 0.436 | 0.448 | 0.388 | 0.419 |
| ETTh2 720 | | 0.409 | 0.447 | 0.387 | 0.429 | 0.803 | 0.633 | 0.417 | 0.456 | 0.414 | 0.449 | 0.405 | 0.440 | 0.402 | 0.448 | 0.390 | 0.427 | 0.451 | 0.473 | 0.406 | 0.438 | 0.473 | 0.482 | 0.408 | 0.443 |
| ETTh2 *Avg.* | | 0.367 | 0.411 | 0.349 | 0.391 | 0.500 | 0.480 | 0.363 | 0.407 | 0.361 | 0.402 | 0.348 | 0.387 | 0.389 | 0.435 | 0.379 | 0.416 | 0.370 | 0.414 | 0.362 | 0.400 | 0.406 | 0.430 | 0.364 | 0.403 |
| Traffic 96 | | 0.551 | 0.362 | 0.497 | 0.337 | 0.497 | 0.361 | 0.528 | 0.368 | 0.894 | 0.609 | 0.662 | 0.469 | 0.516 | 0.342 | 0.490 | 0.332 | 0.677 | 0.474 | 0.582 | 0.406 | 0.993 | 0.635 | 0.601 | 0.434 |
| Traffic 192 | | 0.545 | 0.360 | 0.499 | 0.337 | 0.508 | 0.365 | 0.537 | 0.370 | 1.030 | 0.651 | 0.723 | 0.479 | 0.556 | 0.392 | 0.518 | 0.347 | 1.414 | 0.785 | 0.615 | 0.430 | 0.676 | 0.459 | 0.638 | 0.445 |
| Traffic 336 | | 0.546 | 0.359 | 0.506 | 0.339 | 0.522 | 0.371 | 0.549 | 0.373 | 0.984 | 0.625 | 0.668 | 0.460 | 0.599 | 0.403 | 0.543 | 0.358 | 1.388 | 0.776 | 0.639 | 0.442 | 0.744 | 0.500 | 0.691 | 0.477 |
| Traffic 720 | | 0.592 | 0.378 | 0.538 | 0.353 | 0.572 | 0.394 | 0.593 | 0.392 | 1.020 | 0.631 | 0.821 | 0.535 | 0.692 | 0.447 | 0.605 | 0.387 | 1.599 | 0.851 | 0.707 | 0.473 | 0.778 | 0.511 | 0.757 | 0.501 |
| Traffic *Avg.* | | 0.559 | 0.365 | 0.510 | 0.342 | 0.525 | 0.373 | 0.552 | 0.376 | 0.982 | 0.629 | 0.719 | 0.486 | 0.591 | 0.396 | 0.539 | 0.356 | 1.270 | 0.721 | 0.636 | 0.438 | 0.798 | 0.526 | 0.672 | 0.464 |
| Electricity 96 | | 0.217 | 0.300 | 0.157 | 0.255 | 0.184 | 0.302 | 0.157 | 0.261 | 0.330 | 0.431 | 0.164 | 0.276 | 0.198 | 0.283 | 0.169 | 0.264 | 0.197 | 0.315 | 0.172 | 0.287 | 0.202 | 0.319 | 0.153 | 0.260 |
| Electricity 192 | | 0.228 | 0.308 | 0.176 | 0.274 | 0.191 | 0.305 | 0.168 | 0.271 | 0.214 | 0.326 | 0.187 | 0.300 | 0.235 | 0.331 | 0.196 | 0.288 | 0.215 | 0.335 | 0.178 | 0.287 | 0.230 | 0.344 | 0.173 | 0.279 |
| Electricity 336 | | 0.246 | 0.325 | 0.196 | 0.292 | 0.208 | 0.321 | 0.185 | 0.288 | 0.370 | 0.455 | 0.209 | 0.318 | 0.238 | 0.326 | 0.208 | 0.299 | 0.220 | 0.332 | 0.206 | 0.317 | 0.277 | 0.384 | 0.202 | 0.311 |
| Electricity 720 | | 0.263 | 0.338 | 0.220 | 0.314 | 0.219 | 0.325 | 0.218 | 0.317 | 0.644 | 0.613 | 0.268 | 0.380 | 0.268 | 0.348 | 0.241 | 0.329 | 0.337 | 0.426 | 0.256 | 0.362 | 0.365 | 0.451 | 0.250 | 0.348 |
| Electricity *Avg.* | | 0.239 | 0.318 | 0.187 | 0.284 | 0.201 | 0.313 | 0.182 | 0.284 | 0.389 | 0.456 | 0.207 | 0.318 | 0.235 | 0.322 | 0.203 | 0.295 | 0.242 | 0.352 | 0.203 | 0.313 | 0.268 | 0.375 | 0.194 | 0.299 |

These results suggest that the necessity of a well-formed, stationary latent space are robust across representative forecasting backbones. Overall, the appendix results reinforce our conclusion that freezing a pre-trained AutoEncoder and training the forecasting backbone solely with $\mathcal{L}_{\text{Pred}}$ and $\mathcal{L}_{\text{Align}}$ provides the most stable and effective learning paradigm.

## G. Latent-Space Dynamical Structure: Full Metrics

This appendix complements the main-paper summary in App. 5.6 with the full quantitative metrics that characterize the latent-space dynamical structure across datasets. For each dataset we report three complementary metrics: **CKA** (Centered Kernel Alignment between observation and latent representations; values close to 1.0 would indicate identity mapping), **Effective Rank** of the latent covariance spectrum (a higher value indicates more significant directions in the representation), and **Temporal Transition Consistency (TTC)** (cosine similarity between consecutive states; higher values indicate more coherent latent trajectories). The results in Tab. 11 confirm a consistent pattern: the latent space encodes structured dynamics that are quantitatively distinct from a relabeling of the observation space.

*Table 8.* **AE learning-rate grid: iTransformer on ETTh1.** Avg. MSE/MAE over $T \in \{96, 192, 336, 720\}$ under a grid of encoder LR (columns) and AE-decoder LR (rows). **Top:** load a pre-trained AE and fine-tune it. **Bottom:** no pre-trained AE; use 2-layer MLP (AE-style init).

| Model | Dataset | Enc Load | Dec Load | AE Dec LR | Enc LR=0 | | Enc LR=1e-5 | | Enc LR=5e-5 | | Enc LR=1e-4 | | Enc LR=0.001 | |
|---|---|---|---|---|---|---|---|---|---|---|---|---|---|---|
| | | | | | MSE | MAE | MSE | MAE | MSE | MAE | MSE | MAE | MSE | MAE |
| iTransformer | ETTh1 ( 0.441 / 0.459 ) | ✓ | ✓ | 0 | 0.660 | 0.580 | 0.660 | 0.580 | 0.661 | 0.580 | 0.663 | 0.581 | 0.816 | 0.674 |
| | | | | 1e-5 | 0.663 | 0.583 | 0.663 | 0.582 | 0.664 | 0.583 | 0.672 | 0.587 | 0.816 | 0.676 |
| | | | | 5e-5 | 0.693 | 0.604 | 0.673 | 0.591 | 0.673 | 0.591 | 0.681 | 0.595 | 0.828 | 0.686 |
| | | | | 1e-4 | 0.703 | 0.611 | 0.704 | 0.612 | 0.705 | 0.612 | 0.687 | 0.599 | 0.833 | 0.692 |
| | | | | 0.001 | 0.806 | 0.672 | 0.806 | 0.671 | 0.805 | 0.670 | 0.806 | 0.671 | 0.849 | 0.698 |
| | ETTh1 ( 0.441 / 0.459 ) | ✗ | ✗ | 0 | 0.896 | 0.740 | 0.937 | 0.764 | 0.888 | 0.732 | 0.940 | 0.762 | 0.928 | 0.740 |
| | | | | 1e-5 | 0.909 | 0.749 | 0.897 | 0.743 | 0.832 | 0.706 | 0.959 | 0.774 | 0.898 | 0.726 |
| | | | | 5e-5 | 0.895 | 0.739 | 0.878 | 0.730 | 0.832 | 0.716 | 0.962 | 0.773 | 0.906 | 0.732 |
| | | | | 1e-4 | 0.883 | 0.733 | 0.892 | 0.738 | 0.843 | 0.720 | 0.899 | 0.740 | 0.891 | 0.734 |
| | | | | 0.001 | 0.894 | 0.746 | 0.923 | 0.761 | 0.934 | 0.761 | 0.914 | 0.757 | 0.897 | 0.730 |

*Table 9.* **AE learning-rate grid: DLinear on Electricity.** Avg. MSE/MAE over $T \in \{96, 192, 336, 720\}$ under a grid of encoder LR (columns) and AE-decoder LR (rows). **Top:** load a pre-trained AE and fine-tune it. **Bottom:** no pre-trained AE; use 2-layer MLP (AE-style init).

| Model | Dataset | Enc Load | Dec Load | AE Dec LR | Enc LR=0 | | Enc LR=1e-5 | | Enc LR=5e-5 | | Enc LR=1e-4 | | Enc LR=0.001 | |
|---|---|---|---|---|---|---|---|---|---|---|---|---|---|---|
| | | | | | MSE | MAE | MSE | MAE | MSE | MAE | MSE | MAE | MSE | MAE |
| DLinear | Electricity ( 0.182 / 0.284 ) | ✓ | ✓ | 0 | 0.284 | 0.381 | 0.280 | 0.379 | 0.279 | 0.376 | 0.277 | 0.375 | 0.290 | 0.386 |
| | | | | 1e-5 | 0.271 | 0.369 | 0.271 | 0.371 | 0.275 | 0.373 | 0.287 | 0.386 | 0.288 | 0.384 |
| | | | | 5e-5 | 0.267 | 0.366 | 0.268 | 0.367 | 0.265 | 0.365 | 0.265 | 0.365 | 0.293 | 0.388 |
| | | | | 1e-4 | 0.277 | 0.378 | 0.278 | 0.380 | 0.270 | 0.371 | 0.267 | 0.368 | 0.286 | 0.381 |
| | | | | 0.001 | 0.308 | 0.392 | 0.309 | 0.392 | 0.309 | 0.395 | 0.309 | 0.394 | 0.371 | 0.442 |
| | Electricity ( 0.182 / 0.284 ) | ✗ | ✗ | 0 | 0.330 | 0.420 | 0.328 | 0.420 | 0.325 | 0.419 | 0.331 | 0.423 | 0.359 | 0.458 |
| | | | | 1e-5 | 0.332 | 0.423 | 0.329 | 0.420 | 0.326 | 0.419 | 0.332 | 0.423 | 0.341 | 0.438 |
| | | | | 5e-5 | 0.331 | 0.421 | 0.331 | 0.422 | 0.327 | 0.420 | 0.328 | 0.421 | 0.340 | 0.437 |
| | | | | 1e-4 | 0.331 | 0.421 | 0.330 | 0.422 | 0.330 | 0.422 | 0.329 | 0.420 | 0.341 | 0.439 |
| | | | | 0.001 | 0.342 | 0.438 | 0.337 | 0.434 | 0.338 | 0.435 | 0.339 | 0.436 | 0.372 | 0.464 |

# H. Decoupling Latent Prediction and Alignment Loss

Tab. 12 reports the full decoupling ablation summarized in App. 5.5. The four configurations are: (a) full LatentTSF, (b) LatentTSF with $\mathcal{L}_{\text{Align}}$ removed (so training uses only $\mathcal{L}_{\text{Pred}}$ in the latent space), (c) the DLinear baseline trained with $\mathcal{L}_{\text{Align}}$ applied directly on the observation space (i.e., aligning $\widehat{\mathbf{Y}}$ with $\mathbf{Y}$), and (d) the unmodified DLinear baseline. The ranking $(a) > (b) > (c) \approx (d)$ holds on all five datasets, demonstrating that latent-space prediction is the primary contributor and $\mathcal{L}_{\text{Align}}$ provides additional refinement only when the latent space is well-structured.

# I. Encoder Pre-Training Sensitivity

A natural concern with the frozen-encoder design is sensitivity to the quality of the pre-trained AutoEncoder. We evaluate this by varying the encoder pre-training budget on three datasets, while keeping all backbone training settings fixed. Tab. 13 reports MSE averaged over four prediction lengths. The maximum performance variation across pre-training budgets is only $1.4\%$ (ETTh1), $2.1\%$ (ETTh2), and $8.2\%$ (Electricity), and *every* configuration outperforms the observation-space baseline. Interestingly, an early-stopped encoder (10% epochs) is slightly preferable on ETT, suggesting that the basic cross-channel structure useful for forecasting is captured quickly while prolonged reconstruction training may over-fit to details that are not relevant to downstream forecasting. These results indicate that LatentTSF is robust to encoder quality and does not require careful tuning of the encoder budget.

*Table 10.* **AE learning-rate grid: DLinear on ETTh1.** Avg. MSE/MAE over $T \in \{96, 192, 336, 720\}$ under a grid of encoder LR (columns) and AE-decoder LR (rows). **Top:** load a pre-trained AE and fine-tune it. **Bottom:** no pre-trained AE; use 2-layer MLP (AE-style init).

| Model | Dataset | Enc Load | Dec Load | Dec LR | Enc LR=0 | | Enc LR=1e-5 | | Enc LR=5e-5 | | Enc LR=1e-4 | | Enc LR=0.001 | |
|---|---|---|---|---|---|---|---|---|---|---|---|---|---|---|
| | | | | | MSE | MAE | MSE | MAE | MSE | MAE | MSE | MAE | MSE | MAE |
| DLinear | ETTh1 (0.411 / 0.428) | ✓ | ✓ | 0 | 0.623 | 0.596 | 0.624 | 0.596 | 0.627 | 0.598 | 0.631 | 0.600 | 0.639 | 0.605 |
| | | | | 1e-5 | 0.624 | 0.596 | 0.624 | 0.596 | 0.629 | 0.599 | 0.631 | 0.601 | 0.639 | 0.605 |
| | | | | 5e-5 | 0.626 | 0.597 | 0.627 | 0.598 | 0.628 | 0.599 | 0.631 | 0.600 | 0.645 | 0.608 |
| | | | | 1e-4 | 0.627 | 0.598 | 0.627 | 0.598 | 0.629 | 0.599 | 0.631 | 0.600 | 0.645 | 0.608 |
| | | | | 0.001 | 0.629 | 0.600 | 0.630 | 0.600 | 0.631 | 0.601 | 0.632 | 0.602 | 0.647 | 0.610 |
| | ETTh1 (0.411 / 0.428) | ✗ | ✗ | 0 | 0.717 | 0.642 | 0.712 | 0.637 | 0.708 | 0.634 | 0.710 | 0.636 | 0.704 | 0.632 |
| | | | | 1e-5 | 0.711 | 0.637 | 0.708 | 0.634 | 0.703 | 0.631 | 0.710 | 0.635 | 0.703 | 0.631 |
| | | | | 5e-5 | 0.707 | 0.634 | 0.706 | 0.633 | 0.705 | 0.632 | 0.707 | 0.634 | 0.700 | 0.629 |
| | | | | 1e-4 | 0.710 | 0.636 | 0.709 | 0.635 | 0.704 | 0.632 | 0.709 | 0.635 | 0.703 | 0.631 |
| | | | | 0.001 | 0.707 | 0.634 | 0.707 | 0.634 | 0.702 | 0.631 | 0.706 | 0.634 | 0.701 | 0.629 |

*Table 11.* **Latent-space dynamical structure across datasets.** For each dataset we compare the original observation space against the LatentTSF latent space along three axes: (i) **CKA** between observation and latent representations measures structural similarity; values far below 1.0 rule out the latent space being an identity copy of the observation space. (ii) **Effective Rank** of the covariance spectrum captures the number of significant directions in the representation; consistent increases (especially the $4.4\times$ rise on Electricity) confirm a non-trivial, more expressive latent representation. (iii) **Temporal Transition Consistency (TTC)** reports the cosine similarity between consecutive states; the $\sim 7\%$ improvement reflects more coherent latent trajectories. Together, these metrics provide quantitative evidence that the latent space encodes structured dynamics, complementing the qualitative t-SNE and frequency-domain visualizations in Figure 1 and Figure 5.

| Dataset | Space | CKA ↓ | Eff. Rank ↑ | TTC ↑ |
|---|---|---|---|---|
| ETTh1 | Observation | – | 2.86 | 0.913 |
| | **Latent** | 0.015 | **3.36** | **0.983** |
| Electricity | Observation | – | 7.89 | 0.894 |
| | **Latent** | 0.023 | **34.90** | **0.967** |

# J. Robustness to Noise and Missing Data

Tab. 14 provides the full robustness sweep summarized in App. 5.7. We evaluate two types of input perturbations on ETTh1: additive Gaussian noise with $\sigma \in \{0, 0.1, 0.2, 0.5\}$, and random missing values with rates in $\{0\%, 10\%, 20\%, 30\%\}$. For both DLinear and PatchTST backbones, LatentTSF maintains a lower MSE than the corresponding observation-space training at every corruption level. The improvement is most pronounced under heavy corruption (e.g., PatchTST under $30\%$ missing: $0.5044 \rightarrow 0.4761$), which we interpret as evidence that the latent space encodes information that goes beyond fitting low-order statistical patterns of the clean data.

# K. Latent Dimension Sweep

To support the dimension-agnostic phrasing in the abstract and App. 3, we report a latent-dimension sweep on Traffic ($C = 862$ channels) using the PatchTST backbone. Tab. 15 shows that LatentTSF outperforms the observation-space baseline at both $D < C$ (compression) and $D = C$. This confirms that the framework is effective regardless of whether the latent space is expanded or compressed, supporting the view that the core idea is latent-state modeling rather than feature expansion per se. For high-dimensional channel sets, a compact latent space ($D = 512$) is therefore both memory-efficient and more effective than the original observation space. Settings with $D > 862$ exceeded GPU memory in our setup and are not reported.

# L. Limitations and Future Work

We highlight three limitations of LatentTSF that point to natural directions for future work.

*Table 12.* **Decoupling latent-space prediction from $\mathcal{L}_{\textbf{Align}}$.** DLinear backbone, MSE averaged over four prediction lengths $\{96, 192, 336, 720\}$. The ranking $(a) > (b) > (c) \approx (d)$ holds on every dataset, supporting two conclusions: (i) latent-space prediction is the primary contribution, since moving from observation to latent space (b vs. d) already yields most of the gain (e.g., Electricity 0.1830 vs. 0.2006, ↓ 8.8%); (ii) $\mathcal{L}_{\text{Align}}$ provides additional refinement only when applied in a well-structured latent space, while applying it on the observation space (c) is inconsistent and can even degrade performance (ETTh2: 0.5063 vs. 0.5004). The two design choices are therefore complementary, not competing.

| Configuration | ETTh1 | ETTh2 | ETTm1 | ETTm2 | Electricity |
|---|---|---|---|---|---|
| (a) LatentTSF (full) | **0.4197** | **0.4225** | **0.3543** | **0.2582** | **0.1820** |
| (b) LatentTSF w/o $\mathcal{L}_{\text{Align}}$ | 0.4222 | 0.4311 | 0.3562 | 0.2634 | 0.1830 |
| (c) DLinear $+ \mathcal{L}_{\text{Align}}$ on observation space | 0.4274 | 0.5063 | 0.3548 | 0.2595 | 0.1960 |
| (d) DLinear (baseline) | 0.4275 | 0.5004 | 0.3578 | 0.2661 | 0.2006 |

*Table 13.* **Encoder pre-training sensitivity.** DLinear backbone, MSE averaged over four prediction lengths. The maximum performance variation across encoder pre-training budgets is only 1.4% (ETTh1), 2.1% (ETTh2), and 8.2% (Electricity), and every configuration outperforms the observation-space baseline. The slight advantage of early-stopped encoders on ETT suggests that the basic cross-channel structure useful for forecasting is captured quickly, while prolonged reconstruction training may overfit to details less relevant to forecasting. These results indicate that LatentTSF is robust to encoder quality and that a well-formed latent space provides consistent value as long as the encoder is reasonably trained.

| Encoder pre-training | ETTh1 MSE | ETTh2 MSE | Electricity MSE |
|---|---|---|---|
| 10% epochs (50 ep) | **0.4157** | **0.3926** | **0.1914** |
| 30% epochs (150 ep) | 0.4193 | 0.3998 | 0.1954 |
| 60% epochs (300 ep) | 0.4204 | 0.4009 | 0.2071 |
| 100% epochs (500 ep) | 0.4216 | 0.3989 | 0.1951 |
| *Observation-space baseline (DLinear)* | *0.4275* | *0.5004* | *0.2006* |

*Table 14.* **Robustness under input corruption (ETTh1, MSE averaged over four prediction lengths).** *Top:* additive Gaussian noise with standard deviation $\sigma$ applied to inputs. *Bottom:* random missing values at the input. Across both perturbation types and both backbones, LatentTSF maintains a lower MSE than the corresponding observation-space training at every corruption level. The latent-space training appears to encode information that generalizes beyond fitting clean low-order statistical patterns, supporting the claim that LatentTSF learns more robust representations of the underlying dynamics.

| Noise level $\sigma$ | DLinear | | PatchTST | |
|---|---|---|---|---|
| | Original | w/ LatentTSF | Original | w/ LatentTSF |
| 0 (clean) | 0.4275 | **0.4197** | 0.4366 | **0.4174** |
| 0.1 | 0.4279 | **0.4202** | 0.4374 | **0.4171** |
| 0.2 | 0.4291 | **0.4218** | 0.4401 | **0.4170** |
| 0.5 | 0.4379 | **0.4323** | 0.4624 | **0.4266** |

| Missing rate | DLinear | | PatchTST | |
|---|---|---|---|---|
| | Original | w/ LatentTSF | Original | w/ LatentTSF |
| 0% | 0.4275 | **0.4197** | 0.4366 | **0.4174** |
| 10% | 0.4453 | **0.4369** | 0.4490 | **0.4261** |
| 20% | 0.4713 | **0.4635** | 0.4698 | **0.4443** |
| 30% | 0.5125 | **0.5057** | 0.5044 | **0.4761** |

*Table 15.* **Effect of latent dimension on Traffic.** PatchTST backbone, MSE averaged over four prediction lengths; $C = 862$ is the original number of channels. LatentTSF outperforms the observation-space baseline at both $D < C$ (compression) and $D = C$, demonstrating that the framework is effective regardless of whether the latent space is expanded or compressed relative to the observation space. The core idea is therefore latent-state modeling rather than feature expansion per se: for high-dimensional channel sets, a compact latent space ($D = 512$) is both memory-efficient and more effective than the original observation space. Larger settings ($D > 862$) exceeded GPU memory in our setup and are not reported.

| Latent dim. $D$ | Relation to $C = 862$ | LatentTSF MSE | Baseline MSE |
|---|---|---|---|
| 512 | $D < C$ (compression) | **0.786** | 0.982 |
| 862 | $D = C$ | **1.084** | 1.297 |

**(i) Unstructured latent operator.** The latent transition function $\mathcal{F}_\theta^{\mathbf{Z}}$ is intentionally left unstructured to keep LatentTSF backbone-agnostic. Imposing explicit operator structure (e.g., Koopman-style spectral or stability constraints) is a promising extension that may further improve long-horizon stability at the cost of generality.

**(ii) Frozen encoder.** Our framework relies on a pre-trained, frozen point-wise AutoEncoder. Encoder-quality sensitivity is empirically bounded (App. I), but truly learnable joint encoder–dynamics training that preserves the stationary-target property remains open.

**(iii) Surrogate, not strict bound.** Our final $\mathcal{L}_{\text{Align}}$ is a non-contrastive cosine surrogate inspired by mutual information maximization rather than a strict variational lower bound; deriving a fully rigorous bound that retains the empirical benefits of the non-contrastive form is left to future work.

