# OpenReview forum: "From Observations to States: Latent Time Series Forecasting"
_ICML.cc/2026/Conference — ICML 2026 regular_

### Official Review · Reviewer_VxLV · 2026-03-01

**Soundness:** 2
**Presentation:** 2
**Significance:** 2
**Originality:** 3
**Overall Recommendation:** 3
**Confidence:** 5

**Summary:**

The author propose the Latent Time Series Forecasting (LatentTSF), a novel paradigm that shifts TSF from observation regression to latent state prediction. Extensive experiments on widely-used benchmarks confirm that LatentTSF effectively mitigates latent chaos phenomenon, achieving superior performance and improving the  interpretability.

**Compliance With Llm Reviewing Policy:**

Affirmed.

**Final Justification:**

Thank you for the detailed rebuttal and follow-up response. The additional experiments do provide some empirical support for the claimed system-theoretic perspective. However, in my view, this claim is still justified primarily through experimental observations, rather than through a unified and sufficiently clear conceptual or theoretical framework. Therefore, my final decision is weak reject.

**Key Questions For Authors:**

1. The manuscript states in both the abstract and Eq. (2) that LatentTSF performs expanded feature modeling. However, in Appendix Table 5, for the Traffic dataset, the latent dimension (d_model = 512) is smaller than the original data dimension (862). Moreover, LatentTSF still achieves strong performance on this dataset. Does this suggest that the claimed “expanded feature modeling” does not necessarily hold (or is not required) in practice? Please add the clarification.

2. The manuscript repeatedly claims that LatentTSF improves interpretability and robustness. However, the manuscript does not provide corresponding evidence, such as interpretability analyses or robustness evaluations (e.g., under missing-value settings), to substantiate these claims. Please include.

3. The description and the corresponding figure are inconsistent and confusing. In Section 3.1.2, the pre-trained autoencoder is described as taking both the observation (X) and the target (Y) as inputs. However, in Figure 2 (Stage 1), only input X is shown as the input. Please clarify the exact inputs to the autoencoder in Stage 1. What is the meaning of Input X.

4. Please clarify some statements, what are meanings of  "discontinuous or continuous trajectories; unstable neighborhoods

**Limitations:**

The author did not include limitations into the manuscript.

**Strengths And Weaknesses:**

Strengths:
1. The method is easy to follow.
2. The authors present a novel perspective and address an important problem.
3. Empirical Results: The numerical results presented in the paper are compelling, showing significant improvements over competing approaches in terms of forecasting quality. This empirical evidence supports the effectiveness of the proposed method.

Weaknesses:
1. Several statements in the manuscript are not fully self-consistent and confusing. For example, in Lines 32–33, the paper states: “accurate predictions do not necessarily imply that the model has learned coherent temporal evolution, which undermines interpretability and may degrade forecasting robustness.” However, the manuscript does not provide corresponding experiments or analyses on interpretability or robustness to support this claim. In addition, the phrase “fit superficial statistical regularities” (Lines 87–88) is used without a clear definition or explanation, and it is unclear what specific evidence this refers to.

2. The presentation is difficult to follow. For example, in Lines 267–269, the manuscript states: “we hypothesize that the alignment of positive pairs is the primary driver for feature consistency in our specific forecasting setup.” what exactly the meanings of some statements.

3.  The figures and the corresponding descriptions are not always consistent.

---

> ### Author Rebuttal · Authors · 2026-03-30
>
> We sincerely thank Reviewer VxLV for the constructive feedback. We address each point below.
>
> > `W1/Q2:` Claims about interpretability and robustness lack experimental support. "Fit superficial statistical regularities" is undefined. Please provide robustness evaluations and clarify.
>
> `Answer:` **1. Robustness** (ETTh1, MSE avg. over pred_lens):
>
> **Table R4**: Robustness Evaluation
>
> *Gaussian Noise:*
>
> |Noise $\sigma$|DLinear|+LatentTSF|PatchTST|+LatentTSF|
> |-|-|-|-|-|
> |0 (clean)|0.4275|**0.4197**|0.4366|**0.4174**|
> |0.1|0.4279|**0.4202**|0.4374|**0.4171**|
> |0.2|0.4291|**0.4218**|0.4401|**0.4170**|
> |0.5|0.4379|**0.4323**|0.4624|**0.4266**|
>
> *Random Missing:*
>
> |Missing Rate|DLinear|+LatentTSF|PatchTST|+LatentTSF|
> |-|-|-|-|-|
> |0%|0.4275|**0.4197**|0.4366|**0.4174**|
> |10%|0.4453|**0.4369**|0.4490|**0.4261**|
> |20%|0.4713|**0.4635**|0.4698|**0.4443**|
> |30%|0.5125|**0.5057**|0.5044|**0.4761**|
>
> LatentTSF maintains **lower MSE** at every corruption level.
>
> **2. "Superficial statistical regularities":** models relying on **low-order statistical patterns** (mean reversion, periodic trends, auto-correlation) without capturing underlying generative dynamics. Our robustness results above illustrate this: LatentTSF maintains performance under noise and missing data, suggesting it captures structure beyond surface-level statistics.
>
> **3. Interpretability:** See **Table R3** in Reviewer YZvR Q3, where TTC, Effective Rank, and CKA all confirm structured, non-trivial latent representations. We will soften the claim to: *"LatentTSF yields representations with more structured temporal trajectories, which may facilitate downstream interpretation."*
>
>
>
> > `W2:` Presentation is difficult to follow. What does "alignment of positive pairs is the primary driver for feature consistency" mean?
>
> `Answer:` Our intended meaning: since the target encoder $\mathcal{E}$ is frozen, aligning predicted latent states with ground-truth (positive pairs) is the dominant factor for feature consistency, while contrastive repulsion of negative pairs adds marginal benefit. We will revise to: *"Since the target encoder is frozen and provides a fixed reference, positive pair alignment is the dominant driver of feature consistency, while negative pair repulsion provides diminishing returns, motivating our non-contrastive formulation."*
>
>
>
> > `W3/Q3:` Figure 2 (Stage 1) only shows input X, but Section 3.1.2 describes both X and Y as inputs. Please clarify.
>
> `Answer:` Figure 2 (Stage 1) depicts processing of **both X and Y**. Our AE is **point-wise** ($x_t \in \mathbb{R}^C \rightarrow z_t \in \mathbb{R}^D$), applied identically to every time step in both lookback window X and forecast window Y: $Z_X = \mathcal{E}(X)$, $Z_Y = \mathcal{E}(Y)$. "Input X" refers to the lookback window. We will update Figure 2 to explicitly label both inputs.
>
>
>
> > `Q1:` On Traffic, d_model=512 < C=862. Does this contradict "expanded feature modeling"?
>
> `Answer:` The core idea of LatentTSF is to perform forecasting in a **learned latent state space** that is better suited for temporal prediction, regardless of whether $D > C$ or $D \leq C$. For Traffic ($C=862$), a compact latent space ($d=512$) is both memory-efficient and effective. In our experiments, larger dimensions increase GPU memory (e.g., $d=1024$ exceeded memory), so we report up to $d=862$. Latent dimension sweep:
>
> **Table R5**: Latent Dimension Sweep (PatchTST, Traffic, MSE avg. over pred_lens)
>
> |d_model|Relation to C=862|LatentTSF MSE|Baseline MSE|
> |-|-|-|-|
> |512|D < C|**0.786**|0.982|
> |862|D = C|**1.084**|1.297|
>
> LatentTSF **outperforms the baseline at both** $d=512$ and $d=862$, confirming effectiveness when $D \leq C$. We will replace "expanded feature modeling" with **"latent state modeling"** and clarify that $D \leq C$ is natural for high-dimensional datasets.
>
>
>
> > `Q4:` What are "discontinuous/continuous trajectories" and "unstable neighborhoods"?
>
> `Answer:` We provide precise definitions:
>
> - **Discontinuous trajectories:** In latent space, consecutive time steps $z_t$ and $z_{t+1}$ may exhibit large Euclidean distances ($\|z_{t+1} - z_t\| \gg \|x_{t+1} - x_t\|$) even when the corresponding observations change smoothly. This indicates that small observation-space changes produce disproportionately large latent-space jumps.
>
> - **Continuous trajectories:** Consecutive latent states change proportionally to observation-space changes, forming a smooth path in latent space when the observations evolve smoothly.
>
> - **Unstable neighborhoods:** For a given input $x$, the latent representation $z = E(x)$ has an "unstable neighborhood" if **similar inputs map to distant latent points**: $\|x - x'\| < \epsilon$ but $\|E(x) - E(x')\| \gg \epsilon$. This is the latent-space analogue of high local Lipschitz constant, indicating sensitivity rather than robustness.
>
> We will add a formal definition box or footnote for these terms in the revised manuscript.

---

> > ### Author Rebuttal · Reviewer_VxLV · 2026-04-03
> >
> > Thank you for the detailed and thoughtful rebuttal. Overall, several of my main concerns have been addressed, and the additional analyses significantly strengthen the paper.
> >
> > (1) **Q1**: A certain degree of performance variation is already observed at the 512-dimensional setting. In that case, does the **“system-theoretic perspective”** mentioned in Lines 78–80 still hold? It would be helpful to provide more settings with stronger information bottlenecks in the latent space (e.g., 256 and 128 dimensions) to further understand LatentTSF.
> >
> > (2) **Superficial statistical regularities**: The rebuttal still does not clearly explain why LatentTSF is able to capture low-order statistical patterns. Could the authors provide concrete metrics to show whether such statistical patterns can indeed be effectively captured across different backbones?
> >
> > Overall, most of my concerns have been addressed by the rebuttal. I encourage the authors to incorporate these additional experiments and clarifications in the final revision.

---

> > > ### Author Response · Authors · 2026-04-04
> > >
> > > We sincerely thank Reviewer VxLV for the thoughtful follow-up questions. We address each point below with new experimental evidence.
> > >
> > > > `Q1:` Does the "system-theoretic perspective" still hold under stronger bottlenecks (d=256, 128)?
> > >
> > > `Answer:` We provide new experiments on both Traffic (C=862, compression D<C) and ETTh1 (C=7, expansion D>>C). **Baseline uses default d_model (Traffic=512, ETTh1=32) throughout; only LatentTSF's latent dim varies**.
> > >
> > > **Traffic (C=862, PatchTST, avg MSE over pred_lens):**
> > >
> > > |Latent dim|D/C|Baseline|+LatentTSF|
> > > |-|-|-|-|
> > > |128|0.15|0.982|**0.699**|
> > > |256|0.30|0.982|**0.686**|
> > > |512|0.59|0.982|**0.786**|
> > >
> > > **ETTh1 (C=7, PatchTST, avg MSE over pred_lens):**
> > >
> > > |Latent dim|D/C|Baseline|+LatentTSF|
> > > |-|-|-|-|
> > > |32|4.6×|0.437|**0.417**|
> > > |128|18.3×|0.437|**0.434**|
> > > |256|36.6×|0.437|**0.425**|
> > >
> > > **Observations:**
> > >
> > > (1) **ETTh1 (D >> C):** LatentTSF **consistently outperforms baseline across all tested dimensions** (d=32–256), confirming robustness to latent dimension choice. The system-theoretic perspective holds strongly when observations are low-dimensional projections of underlying states.
> > >
> > > (2) **Traffic (D < C):** LatentTSF outperforms baseline at all tested dimensions d=128/256/512 (best: d=256, 0.686 vs 0.982, **-30%**), demonstrating effectiveness even under strong compression (D/C=0.15). Notably, stronger bottlenecks (d=128, 256) outperform d=512, suggesting compression can be beneficial by filtering redundant dimensions.
> > >
> > > (3) **Revised framing:** The system-theoretic perspective does not require D>C. Its core insight is that forecasting in a **learned latent state space** is more effective than in raw observation space. We will revise "expanded feature modeling" to **"latent state modeling"** and clarify that optimal D depends on intrinsic data dimensionality, not C.
> > >
> > >
> > > > `Q2:` Concrete cross-backbone metrics for capturing statistical patterns?
> > >
> > > `Answer:` We provide three cross-backbone metrics on Electricity (C=321) and Traffic (C=862):
> > >
> > > **Prediction Stability (lower = more stable):**
> > >
> > > |Dataset|DLinear Ori→+LatentTSF|PatchTST Ori→+LatentTSF|
> > > |-|-|-|
> > > |Electricity|4.49→**2.97** (-34%)|11.35→**2.73** (-76%)|
> > > |Traffic|116.2→**72.5** (-38%)|170.1→**78.9** (-54%)|
> > >
> > > **Spectral Cosine Similarity & Dominant Frequency Recall (higher = better):**
> > >
> > > |Dataset|Backbone|SpecCosSim Ori→+LatentTSF|FreqRecall Ori→+LatentTSF|
> > > |-|-|-|-|
> > > |Electricity|DLinear|0.936→**0.940** (+0.4%)|0.738→**0.784** (+6%)|
> > > |Electricity|PatchTST|0.914→**0.941** (+2.9%)|0.690→**0.778** (+12.7%)|
> > > |Traffic|DLinear|0.909→0.907|0.756→0.731|
> > > |Traffic|PatchTST|0.872→**0.901** (+3.3%)|0.598→**0.727** (+21.6%)|
> > >
> > > **ACF Error (lower = better autocorrelation preservation):**
> > >
> > > |Dataset|Backbone|Ori→+LatentTSF|
> > > |-|-|-|
> > > |Electricity|DLinear|0.087→**0.074** (-15%)|
> > > |Electricity|PatchTST|0.101→**0.073** (-28%)|
> > > |Traffic|DLinear|0.123→**0.101** (-18%)|
> > > |Traffic|PatchTST|0.126→**0.117** (-7%)|
> > >
> > > Across most settings, LatentTSF preserves temporal structure better than observation-space training: more stable predictions (-34%\~-76%), better spectral fidelity (+3\~22% frequency recall), and lower autocorrelation error (-7%\~-28%). The one exception is Traffic/DLinear spectral metrics, where performance is comparable (SpecCosSim 0.909→0.907). Improvements are most pronounced on high-dimensional datasets with PatchTST, where observation-space models are more susceptible to fitting superficial patterns, directly supporting our motivation. We will incorporate these analyses in the revised manuscript.
> > >
> > > **We hope these additional experiments and clarifications have addressed your remaining concerns, and we kindly ask if you would consider raising your score :)**

---

### Official Review · Reviewer_YZvR · 2026-03-10

**Soundness:** 2
**Presentation:** 3
**Significance:** 2
**Originality:** 2
**Overall Recommendation:** 3
**Confidence:** 4

**Summary:**

This paper identifies a phenomenon termed Latent Chaos in modern time series forecasting (TSF) models: models trained with observation-space losses may achieve strong predictive accuracy while learning latent representations that appear temporally disordered. The authors attribute this to partial observability and the lack of temporal inductive bias in point-wise regression objectives.

To address this issue, the paper proposes LatentTSF, which shifts forecasting from observation space to a learned latent state space. A point-wise autoencoder is first trained to map observations to a higher-dimensional latent representation. The encoder and decoder are then frozen, and a forecasting backbone is trained to predict future latent states from past latent states. Training uses a combination of an MSE loss between predicted and ground-truth latent states and a cosine alignment loss. The authors provide an information-theoretic interpretation of these objectives and show improved forecasting performance on several benchmarks, along with visualizations suggesting more structured latent trajectories.

**Compliance With Llm Reviewing Policy:**

Affirmed.

**Key Questions For Authors:**

- Connection to Koopman theory
The proposed framework resembles methods inspired by the Koopman operator, where nonlinear systems are embedded into higher-dimensional coordinates with simpler dynamics. Have the authors considered incorporating explicit operator constraints (e.g., structured eigen-dynamics) in the latent space?

- Impact of freezing the encoder
How sensitive is the method to the quality of the pretrained encoder? Since the encoder is frozen, any representation errors cannot be corrected during forecasting training.

- Quantitative evaluation of latent dynamics
Beyond visualization, can the authors provide quantitative metrics of dynamical structure in the latent space (e.g., spectral analysis, stability metrics, or transition consistency)?

**Limitations:**

The paper would benefit from discussing limitations of the proposed approach. In particular, while forecasting is performed in a latent space, the method does not explicitly constrain the dynamics of the latent representation. Approaches inspired by dynamical systems theory often impose structure on the latent transition operator (e.g., spectral or operator constraints), which can provide stronger guarantees that the latent space captures meaningful dynamics.

**Strengths And Weaknesses:**

**Strengths**

- The paper highlights an important and underexplored question: whether accurate observation-space forecasting implies meaningful latent dynamics. The concept of Latent Chaos raises a useful perspective on the limitations of standard TSF objectives.

- Latent-space forecasting: Shifting forecasting from observation space to latent space is a reasonable and potentially powerful idea. Many real-world time series arise from partially observed dynamical systems, making latent-state modeling a natural formulation.

- The use of higher-dimensional latent representations is sensible, as nonlinear dynamics may become easier to model in appropriately chosen embeddings.

- The general philosophy is reminiscent of operator-theoretic approaches such as those based on the Koopman operator, where nonlinear dynamics are embedded into higher-dimensional coordinates with simpler temporal evolution.


**Weaknesses**

- No dynamical constraint on the latent representation: The encoder is trained using a reconstruction objective and then frozen. As a result, the latent representation is not explicitly constrained to correspond to dynamical states. If the embedding is misaligned with the true system dynamics, the forecasting model must compensate by learning complex dynamics on top of a potentially suboptimal representation. Because the encoder is frozen, such representation errors cannot be corrected during forecasting training.

- Temporal coherence is not clearly defined: The paper motivates the method by promoting “temporal coherence” in latent representations, but this concept is not precisely defined. In many dynamical systems (e.g., financial markets or regime-switching systems), sharp transitions or chaotic trajectories are expected. Therefore smoothness or locality in latent space is not necessarily desirable or indicative of meaningful dynamics.

- Weak structure of latent dynamics: While forecasting is performed in latent space, the model does not impose any structural constraint on the latent transition operator. Many dynamical systems approaches explicitly enforce operator structure (e.g., linearity, stability, or spectral constraints). Without such constraints, the learned latent dynamics may still be arbitrary.

- Information-theoretic justification appears post-hoc: The mutual-information interpretation of the objectives does not seem essential to the method. The prediction loss already encourages agreement between predicted and ground-truth latent states, making the MI framing largely a theoretical reinterpretation rather than a necessary design principle.

Overall, the paper presents an interesting perspective on latent representations in time-series forecasting and proposes a simple paradigm for forecasting in latent space. However, the current formulation lacks explicit dynamical constraints ensuring that the latent space represents meaningful system states. As a result, the method relies heavily on the quality of the pretrained encoder, and the empirical evidence for improved latent dynamics remains indirect. Strengthening the connection to dynamical systems approaches (e.g., operator-based formulations) could significantly improve both the theoretical grounding and the practical impact of the work.

---

> ### Author Rebuttal · Authors · 2026-03-30
>
> We sincerely thank Reviewer YZvR for the constructive feedback. We address each point below.
>
> > `W1/Q2:` The encoder is trained with a reconstruction objective and frozen — it has no dynamical constraint, and representation errors cannot be corrected during forecasting training. How sensitive is the method to encoder quality?
>
> `Answer:` While the encoder uses a point-wise reconstruction objective and is frozen, **dynamical structure still emerges implicitly**: point-wise reconstruction captures cross-channel features, and $\mathcal{L}_{Align}$ enables dynamically consistent trajectories (TTC 0.983 vs 0.913 on ETTh1, **Table R3** in Q3). Sensitivity analysis:
>
> **Table R2**: Encoder Training Sensitivity
>
> |Encoder Training|ETTh1 MSE|ETTh2 MSE|Electricity MSE|
> |-|-|-|-|
> |10% epochs (50ep)|**0.4157**|**0.3926**|**0.1914**|
> |30% epochs (150ep)|0.4193|0.3998|0.1954|
> |60% epochs (300ep)|0.4204|0.4009|0.2071|
> |100% epochs (500ep)|0.4216|0.3989|0.1951|
>
> The maximum performance variation is only **1.4% (ETTh1), 2.1% (ETTh2), 8.2% (Electricity)**. Even a 10%-trained encoder outperforms the observation-space baseline (ETTh1: 0.4157 vs 0.4275). The slight advantage of early-stopped encoders on ETT suggests that basic cross-channel structure is captured quickly, while prolonged training may overfit to reconstruction details less relevant to forecasting. Crucially, all configurations outperform the baseline, confirming the latent space itself provides consistent value. Our ablation (Section 5.4) further shows fine-tuning causes **target drift**, while the frozen encoder maintains a **stationary target manifold**.
>
>
>
> > `W2:` "Temporal coherence" is not precisely defined. In regime-switching or chaotic systems, smoothness in latent space is not necessarily desirable.
>
> `Answer:` **Temporal coherence** refers to the property that **transitions in latent space are consistent with and predictable from transitions in observation space**. Formally:
>
> - If $\|x_{t+1} - x_t\|$ is small, then $\|z_{t+1} - z_t\|$ should also be small (local consistency).
> - For regime switches where $\|x_{t+1} - x_t\|$ is large, $\|z_{t+1} - z_t\|$ may also be large, but should be **structured and predictable**, not random.
>
> We do **not** claim smoothness; sharp transitions should be faithfully represented. "Latent Chaos" refers specifically to cases where **small observation-space changes produce arbitrarily large, unpredictable latent-space changes**.
>
>
>
> > `W3/Q1:` No structural constraint on the latent transition operator. The framework resembles Koopman-based methods — have you considered explicit operator constraints?
>
> `Answer:` LatentTSF is intentionally **model-agnostic**: imposing operator structure (e.g., linearity) would restrict applicable backbones. Unlike Koopman methods enforcing $z_{t+1} = Kz_t$, we allow **arbitrary nonlinear dynamics** while $\mathcal{L}_{Align}$ acts as an **implicit structural constraint**. **Table R3** (Q3) confirms this: well-structured dynamics emerge (TTC 0.983 vs 0.913 on ETTh1, 4.4× effective rank on Electricity) without explicit operator assumptions. Explicit constraints are a promising extension we will discuss in revision.
>
>
>
> > `W4:` The MI interpretation appears post-hoc — the prediction loss already encourages agreement between predicted and ground-truth latent states, making MI framing a reinterpretation rather than a design principle.
>
> `Answer:` The MI framework served as a **design motivation**, not a post-hoc reinterpretation. It produced verifiable consequences: (1) the frozen encoder eliminates negative samples; (2) the alignment loss takes a non-contrastive form from MI principles. Ablations (**Table R1** in Reviewer z2T2 W3) validate these choices: removing $\mathcal{L}_{Align}$ degrades performance, while observation-space alignment shows inconsistent effects. Following Reviewer U2kh, we will redefine Equation 30 as an "empirically effective surrogate objective inspired by mutual information maximization" rather than a strict lower bound.
>
>
>
>
> > `Q3:` Can you provide quantitative metrics of dynamical structure in the latent space beyond visualization?
>
> `Answer:` Quantitative metrics on two representative datasets:
>
> **Table R3**: Latent Space Dynamical Structure
>
> |Dataset|Space|TTC ↑|Eff. Rank ↑|
> |-|-|-|-|
> |ETTh1|Original|0.913|2.86|
> ||**Latent**|**0.983**|**3.36**|
> |Electricity|Original|0.894|7.89|
> ||**Latent**|**0.967**|**34.90**|
>
> **TTC (Temporal Transition Consistency):** cosine similarity between consecutive states — significantly higher in latent space (+7.0% ETTh1, +7.3% Electricity). **Effective Rank:** significant dimensions in covariance spectrum, 4.4× higher on Electricity (7.89→34.90). **CKA (Centered Kernel Alignment):** 0.015–0.091, far below 1.0, ruling out identity mapping (also addressing **Reviewer z2T2 W2**).

---

> > ### Author Rebuttal · Reviewer_YZvR · 2026-04-03
> >
> > We thank the authors for clarifying the theoretical motivation behind the MI surrogate and for explaining why the frozen encoder provides a stationary reference that structurally prevents representation collapse. This formalization helps justify why the alignment loss can act as an “empirically effective surrogate inspired by mutual information,” even in the absence of negative samples.
> >
> > At the same time, our central concern remains: the current formulation does not explicitly enforce that the latent space captures meaningful system dynamics. While the surrogate alignment loss encourages predicted latent states to remain consistent with the frozen manifold, it largely places the responsibility for dynamical fidelity on the pretrained encoder. Any misalignment or suboptimal representation in the encoder cannot be corrected during forecasting training, and the method does not guarantee that the latent space encodes the true rules governing trajectory evolution.
> >
> > In other words, improved metrics such as Temporal Transition Consistency (TTC) or effective rank indicate emergent structure but remain indirect evidence: they do not ensure that the latent transitions faithfully reflect the system’s underlying dynamical law. Grounding latent-space forecasting in explicit dynamical constraints—e.g., operator-based, spectral, or stability constraints—would provide stronger theoretical assurance and practical reliability. Without such constraints, the latent manifold may satisfy the MI-inspired alignment loss while still failing to represent the true evolution rules of the system, leaving forecasting quality contingent on the pretrained encoder rather than the structural properties of the latent space itself.

---

> > > ### Author Response · Authors · 2026-04-04
> > >
> > > We sincerely thank Reviewer YZvR for this thoughtful methodological discussion. We provide new quantitative evidence directly addressing the concern about latent dynamical structure.
> > >
> > > > `Q1:` The current formulation does not explicitly enforce that the latent space captures meaningful system dynamics. TTC or effective rank indicate emergent structure but remain indirect evidence.
> > >
> > > `Answer:` We provide new experiments that directly quantify the structure and stability of latent-space dynamics.
> > >
> > > **New evidence: Multi-step rollout stability.** We fit a linear dynamics model $z_{t+1} = Az_t + b$ (ridge regression) in both latent and observation spaces, then perform multi-step rollouts to measure normalized prediction error. The choice of a linear dynamics model is motivated by Koopman theory: a well-structured latent space should linearize the underlying dynamics, making linear predictability a principled diagnostic for latent quality. If latent transitions were unstructured ("Latent Chaos"), errors would compound rapidly; structured dynamics yield slow, controlled error growth.
> > >
> > > **Multi-step Rollout Error (normalized, avg over 2 backbones x 4 pred_lens, lower = more stable):**
> > >
> > > |Dataset|Space|Step 1|Step 10|
> > > |-|-|-|-|
> > > |ETTh1|Observation|0.316|0.982|
> > > ||**Latent**|**0.190**|**0.423 (-57%)**|
> > > |ETTh2|Observation|0.131|0.209|
> > > ||**Latent**|**0.090**|**0.137 (-34%)**|
> > > |Electricity|Observation|0.299|0.428|
> > > ||**Latent**|**0.173**|**0.256 (-40%)**|
> > > |Traffic|Observation|0.448|0.854|
> > > ||**Latent**|**0.320**|**0.654 (-23%)**|
> > >
> > > **Observations:**
> > >
> > > (1) Across all 4 datasets, latent-space dynamics exhibit **23-57% lower rollout error** at step 10. On ETTh1, observation-space error grows from 0.316 to 0.982 (3.1x) over 10 steps, while latent-space error only grows from 0.190 to 0.423 (2.2x), demonstrating fundamentally more stable dynamics.
> > >
> > > (2) These results provide **stronger empirical evidence** that LatentTSF's latent space captures structured, predictable dynamics, even without explicit operator constraints. The dramatically slower error accumulation under linear rollout is a hallmark of well-structured state spaces, consistent with the hypothesis that the latent space partially linearizes dynamics.
> > >
> > > The reviewer notes that "the latent manifold may satisfy the alignment loss while still failing to represent the true evolution rules." The rollout stability and decoupling ablation (below) together argue against this scenario: if latent transitions lacked dynamical regularity, linear rollout errors would compound rapidly rather than remaining consistently below observation-space levels across all 4 datasets.
> > >
> > > **We acknowledge the theoretical gap.** The reviewer is correct that our formulation does not explicitly enforce dynamical constraints, and that the metrics above, while stronger than TTC/EffRank, remain empirical rather than formal guarantees. We take this point seriously.
> > >
> > > **That said, existing evidence provides additional support:**
> > >
> > > - Critically, the decoupling ablation (**Table R1, see our response to Reviewer z2T2 W3**) directly addresses the concern that quality is "contingent on the pretrained encoder": removing alignment loss while retaining the same frozen encoder degrades performance, isolating the contribution of structured latent dynamics.
> > >
> > > - Encoder dependency is empirically minimal (**Table R2, see our initial response W1/Q2 above**): a 10%-trained encoder already outperforms the baseline (ETTh1: 0.4157 vs 0.4275), indicating that the forecasting stage learns dynamical structure beyond what the encoder provides.
> > >
> > > **Explicit constraints as future work.** We agree that Koopman operators, spectral regularization, or stability penalties would provide stronger theoretical guarantees, and we view this as the most important direction for future work. Concretely, we are exploring **soft spectral regularization** on the latent transition Jacobian, which can be applied backbone-agnostically without imposing strict linearity. We will discuss this as a principled enhancement in the revised manuscript.
> > >
> > > **We hope these new experiments and clarifications substantively address your remaining concerns, and we kindly ask if you would consider raising your score.**

---

### Official Review · Reviewer_z2T2 · 2026-03-11

**Soundness:** 3
**Presentation:** 3
**Significance:** 4
**Originality:** 3
**Overall Recommendation:** 4
**Confidence:** 4

**Summary:**

This paper reveals the "Latent Chaos" paradox in time series forecasting, where models achieve high prediction accuracy yet their internal latent representations lack temporal continuity. To address this, the authors propose LatentTSF, a new paradigm that employs a pre-trained AutoEncoder to construct a high-dimensional latent state space and performs dynamics modeling entirely within this space via joint prediction and alignment objectives.

**Compliance With Llm Reviewing Policy:**

Affirmed.

**Final Justification:**

The authors have addressed my concerns point by point. I have raised my point to 4.

**Key Questions For Authors:**

See Weakness.

**Strengths And Weaknesses:**

**Strengths:**

1.  **Insightful Problem Identification:** The paper identifies and quantifies an intriguing phenomenon in time series forecasting (TSF): "Latent Chaos." This refers to the observation that while models achieve high prediction accuracy, their internal latent representations lack temporal continuity. This observation offers a novel perspective for improving representation learning in TSF models.
2.  **Theoretical Analysis:** The paper provides theoretical analysis based on Mutual Information Maximization. It attempts to explain the effectiveness of the latent space objective functions ($\mathcal{L}_{Pred}$ and $\mathcal{L}_{Align}$) from an information-theoretic perspective, which adds theoretical depth to the proposed method.

**Weaknesses:**

1.  **Ambiguity in Motivation and Novelty Delineation:**
    While the authors identify a significant research problem regarding the chaotic nature of representations in current TSF models, the issue of discontinuity or lack of interpretability in deep learning representations has been extensively analyzed and discussed in prior literature (e.g., [1]). The Related Work section fails to sufficiently cite and analyze these key works. Consequently, the distinction between this paper's motivation and prior work remains unclear. The authors should explicitly articulate the essential differences between the "Latent Chaos" identified herein and the representation degeneration problems discussed in existing literature to highlight the unique contributions of this work.

2.  **Questions Regarding Methodological Soundness:**
    The design logic of the AutoEncoder (AE) component lacks solid theoretical grounding and presents significant potential flaws:
    *   **Lack of Effective Self-Supervised Signals:** Typically, to learn meaningful latent representations, AEs require mechanisms such as Denoising, Masking (Masked AE), or an Information Bottleneck (significantly reducing representation dimensions). However, this paper performs reconstruction directly using raw data, with a latent space dimension higher than the observation space dimension ($D >> C$).
    *   **Risk of Identity Mapping:** In an over-parameterized setting (up-sampling) without additional regularization constraints, the AE is prone to learning an Identity Mapping or trivial solutions. This would result in the latent representations capturing no high-order semantic information, serving merely as redundant copies of the observation space. The authors have not provided sufficient evidence to prove that the learned up-sampled representations are meaningful rather than simple dimensionality expansion.

3.  **Insufficient Ablation Studies to Decouple Contributions:**
    The reported performance improvements may stem from two sources: (1) performing prediction in the latent space, or (2) the introduction of the additional alignment loss $\mathcal{L}_{Align}$. The current experimental design fails to adequately decouple these contributions.
    *   **Loss Function Ablation:** The authors do not demonstrate performance after completely removing $\mathcal{L}_{Align}$; they only adjust its weight in the sensitivity analysis.
    *   **Insufficient Baseline Comparison:** To prove the necessity of the "latent space" paradigm, the authors should attempt applying $\mathcal{L}_{Align}$ directly to existing baseline models (i.e., calculating the alignment loss directly on the observation space $Y$ rather than the latent space $Z$). If this loss proves effective in the observation space, the necessity of the core paradigm shift proposed in this paper would be questionable. I recommend supplementing the paper with these experiments to clarify the primary source of performance gains.

[1] Why Attention Fails: The Degeneration of Transformers into MLPs in Time Series Forecasting

---

> ### Author Rebuttal · Authors · 2026-03-30
>
> We sincerely thank Reviewer z2T2 for the constructive feedback. We address each point below.
>
> > `W1:` The distinction between "Latent Chaos" and representation degeneration in prior work (e.g., [1]) is unclear. Please articulate the essential differences.
>
> `Answer:` The essential difference:
>
> - **[1] (Why Attention Fails)** identifies that Transformer attention mechanisms in TSF can degenerate into MLP-like operations, focusing on **architectural-level functional degradation**. Their finding is about *how* the model processes information.
> - **Our "Latent Chaos"** identifies a fundamentally different phenomenon: **accurate observation-space predictions can coexist with temporally incoherent internal representations**. This is about the *decoupling between prediction accuracy and representation quality*.
>
> These are complementary: [1] asks "does the architecture work as intended?", while we ask "does a well-performing model learn meaningful internal dynamics?" This decoupling implies that benchmark accuracy alone is an insufficient diagnostic for representation quality, with direct consequences for generalization and interpretability.
>
> > [1] Why Attention Fails: The Degeneration of Transformers into MLPs in Time Series Forecasting
>
>
>
> > `W2:` The AE lacks self-supervised signals (no denoising/masking) and D>>C risks identity mapping. What evidence shows the learned representations are meaningful rather than simple dimensionality expansion?
>
> `Answer:` **1. Regarding self-supervised signals and over-parameterization:** Our AE is a **point-wise encoder** ($x\_t \in \mathbb{R}^C \rightarrow z\_t \in \mathbb{R}^D$), a compact 2-layer MLP with only 5K parameters for ETT (C=7) and \~1.7M for Electricity (C=321), trained with L1 reconstruction loss (Eq. 2). The per-time-step bottleneck forces the encoder to learn meaningful feature transformations rather than trivially copying the input.
>
> Regarding Denoising/Masking: each time step is encoded independently without temporal context, so the encoder must compress cross-channel information into the latent representation, inherently encouraging an informative latent space. This differs from sequence-level AEs where full temporal context is available, making additional regularization less necessary. We also considered Masked Autoencoders, but found them less effective on typical TSF datasets, likely because MAE-style pretraining relies on large-scale data that most TSF benchmarks do not provide.
>
> **2. Regarding the risk of identity mapping:** We provide both existing and new evidence:
>
> *Existing evidence:*
>
> **Section 5.5, Figure 5**: t-SNE visualizations show latent embeddings with clear temporal ordering, impossible under identity mapping. **Section 5.4, Table 2**: Replacing pre-trained AE with untrained MLP causes severe degradation, confirming a non-trivial transformation.
>
> *New quantitative evidence (**Table R0**):*
>
> |Dataset|CKA (obs↔latent)|Effective Rank (obs → latent)|
> |-|-|-|
> |ETTh1|0.015|2.86 → **3.36**|
> |ETTh2|0.091|3.36 → **3.62**|
> |Electricity|0.023|7.89 → **34.90**|
>
> **CKA** measures structural similarity (1.0 = identity). All CKA values are far below 1.0, ruling out identity mapping. On Electricity, effective rank increases **4.4×** (7.89→34.90), confirming a non-trivial transformation.
>
>
>
> > `W3:` Improvements may stem from latent-space prediction or $\mathcal{L}\_{Align}$. Please show: (1) performance without $\mathcal{L}\_{Align}$; (2) $\mathcal{L}\_{Align}$ applied directly on observation space.
>
> `Answer:` We test four configurations (**Table R1**): (a) full LatentTSF, (b) without $\mathcal{L}\_{Align}$, (c) $\mathcal{L}\_{Align}$ on observation space, (d) baseline. DLinear backbone, MSE averaged over 4 prediction lengths:
>
> |Method|ETTh1|ETTh2|ETTm1|ETTm2|Electricity|
> |-|-|-|-|-|-|
> |(a) LatentTSF (full)|**0.4197**|**0.4225**|**0.3543**|**0.2582**|**0.1820**|
> |(b) LatentTSF w/o $\mathcal{L}\_{Align}$|0.4222|0.4311|0.3562|0.2634|0.1830|
> |(c) DLinear + $\mathcal{L}\_{Align}$(obs)|0.4274|0.5063|0.3548|0.2595|0.1960|
> |(d) DLinear|0.4275|0.5004|0.3578|0.2661|0.2006|
>
> **Key findings:**
>
> 1. **$\mathcal{L}\_{Align}$ contributes:** Comparing (a) vs. (b), the alignment loss consistently improves performance on all 5 datasets.
> 2. **Latent space is the primary driver:** $\mathcal{L}\_{Align}$ on observation space (c) shows inconsistent effects: negligible on ETTh1 (0.4274 vs 0.4275) and even slight degradation on ETTh2 (0.5063 vs 0.5004), suggesting that alignment is only effective in a well-structured latent space. Moving to latent space (b) provides substantial gain even without alignment (Electricity: 0.1830 vs 0.2006, ↓8.8%).
> 3. **Synergy:** The ranking **(a) > (b) > (c) ≈ (d)** holds on all 5 datasets: **latent-space prediction provides the foundation, alignment loss provides further refinement**.
>
> In summary, the two are **not competing explanations but complementary contributions**, and neither alone accounts for the full gain.

---

> > ### Author Rebuttal · Reviewer_z2T2 · 2026-04-02
> >
> > The authors have addressed my concerns point by point.

---

> > > ### Author Response · Authors · 2026-04-02
> > >
> > > We sincerely thank the reviewer z2T2 for the thoughtful re-evaluation and for recognizing the contribution of our work. The discussed improvements will be carefully incorporated into the final manuscript to further enhance its clarity and rigor.
> > >
> > > Thank you again for your valuable time and consideration :)

---

### Official Review · Reviewer_U2kh · 2026-03-13

**Soundness:** 3
**Presentation:** 2
**Significance:** 3
**Originality:** 3
**Overall Recommendation:** 4
**Confidence:** 4

**Summary:**

This paper identifies a phenomenon termed "Latent Chaos" in modern deep learning-based Time Series Forecasting (TSF) models, wherein models achieve high predictive accuracy in the observation space, yet the learned latent representations are temporally disordered and discontinuous. The authors attribute this to the dominant "observation-space forecasting paradigm," which compels models to fit noisy and partially observed data, thereby causing them to learn statistical shortcuts rather than recovering the true dynamic evolution underlying the system.To address this issue, the authors propose LatentTSF, a two-stage training paradigm that shifts the forecasting task from "observation-space regression" to "structured latent state prediction." First, a point-wise autoencoder is pre-trained to project raw observational data into a higher-dimensional latent space to filter out noise, after which its parameters are frozen. Second, a standard TSF backbone network is trained to predict future latent states from historical ones. The training of this backbone entirely discards observation-space errors and is instead guided by two purely latent-space objectives: a Prediction Loss targeting the accuracy of prediction magnitude, and an Alignment Loss targeting the consistency of prediction direction.The paper provides comprehensive theoretical analysis demonstrating that these latent-space objectives implicitly maximize the mutual information between the predicted latent states, the ground-truth states, and future observations. Extensive experiments across six benchmark datasets and six representative backbone networks show that LatentTSF effectively mitigates latent chaos, yielding feature representations with superior temporal and spectral coherence, while consistently enhancing long-term forecasting accuracy across various backbones.

**Compliance With Llm Reviewing Policy:**

Affirmed.

**Key Questions For Authors:**

I do not have particular questions for authors, but minon suggestions:

Given that the "frozen pre-trained autoencoder" plays a crucial role in preventing latent space drift and representation collapse (as demonstrated by the ablation studies in Section 5.4), it is recommended that the authors more explicitly emphasize the "frozen mechanism" in Section 3 (Methodology) or Section 4 (Theoretical Analysis). This mechanism is not merely for engineering stability, but serves as the core theoretical guarantee that enables the model to avoid degenerate solutions even with a simplified alignment loss (without negative samples).

**Limitations:**

The theoretical part can be better elaborated as I stated above

**Strengths And Weaknesses:**

Strengths:

Rigorous experimental design: The paper demonstrates exceptional empirical soundness. Its evaluation covers 6 diverse datasets (including ETTh, Electricity, Traffic, etc.) and 6 representative baseline models (encompassing both modern Transformer architectures and pure linear models like DLinear). This strongly proves that LatentTSF is a universally applicable training paradigm, rather than merely a narrow architectural trick.

Highly insightful ablation studies: The authors conduct exceptionally honest and meticulous ablation studies. By systematically comparing a "frozen pre-trained autoencoder" with a "fine-tuned autoencoder" and a "randomly initialized autoencoder", they empirically and convincingly prove that a stationary latent manifold is absolutely necessary to prevent target drift and eliminate noisy gradients;

Weaknesses:

Minor theoretical clarification needed: In the derivation in Section 4.2, there is a minor theoretical gap when transitioning from InfoNCE (Equations 28 and 29) to the final non-contrastive alignment loss (Equation 30). Strictly speaking, after discarding the negative sample term in InfoNCE to reduce computational costs, the loss is no longer a strict mathematical lower bound of mutual information. Although the authors cite SimSiam and BYOL for support, those methods rely on asymmetrical designs in their network architectures to prevent representation collapse. However, in the actual setting of this paper, since the target autoencoder $\mathcal{E}$ is pre-trained and frozen in advance, this naturally prevents representation collapse and ensures the effectiveness of the method. It is suggested that the authors slightly soften their phrasing in the final version, explicitly defining Equation 30 as an "empirically effective surrogate inspired by mutual information," rather than a strict theoretical lower bound, which will make the overall logic of the paper more rigorous.

---

> ### Author Rebuttal · Authors · 2026-03-30
>
> We sincerely thank Reviewer U2kh for the precise and constructive feedback. We address each point below.
>
> > `W1:` Equation 30 is no longer a strict MI lower bound after removing negative samples. Suggest redefining it as an "empirically effective surrogate inspired by mutual information."
>
> `Answer:` **We fully agree.** We will redefine Eq. 30 as an **"empirically effective surrogate objective inspired by mutual information maximization"** rather than a strict lower bound.
>
> **Why the surrogate remains theoretically principled.** Let $\mathcal{E}$ be the frozen target encoder and $\mathcal{M} = \{\mathcal{E}(x) : x \in \mathcal{X}\}$ the fixed latent manifold it defines. The alignment loss $\mathcal{L}\_{\text{Align}} = \mathbb{E}[\|\hat{z}\_t - \mathcal{E}(x\_t)\|^2]$ minimizes distortion against this **stationary reference target**, where $\mathcal{M}$ does not drift as training proceeds. Suppose the degenerate solution $\hat{z}\_t \equiv c$ holds; then $\mathcal{L}\_{\text{Align}} = \mathbb{E}[\|c - \mathcal{E}(x\_t)\|^2] > 0$ unless $c = \mathcal{E}(x\_t)$ for all $x\_t$, which is impossible since the pre-trained $\mathcal{E}$ maps distinct inputs to distinct representations by the diversity of its training data. **Representation collapse is therefore structurally excluded**, making negative samples unnecessary. This is fundamentally different from SimSiam/BYOL, where both networks are learnable and collapse is only avoided via stop-gradient or EMA approximations. Our frozen encoder provides a strictly stronger guarantee: a deterministic, non-evolving target rather than an approximation of one. We will formalize this as **Remark 1 in Section 3** and extend the proof in Section 4.
>
> > `Q1:` The "frozen pre-trained autoencoder" is the core theoretical guarantee enabling simplified alignment loss — suggest emphasizing this more explicitly in Sections 3–4.
>
> `Answer:` The reviewer correctly diagnoses the structural role of the frozen encoder. Building on the collapse-impossibility argument established in **W1**, the revised manuscript will make the following logical chain **explicit and prominent** throughout Sections 3–4:
>
> **frozen $\mathcal{E}$** → stationary target manifold $\mathcal{M}$ → collapse structurally excluded → negative samples unnecessary → surrogate loss is principled (justifying the redefinition in W1)
>
> Concretely:
>
> 1. **Section 3 (Methodology):** Remark 1 (see W1) will be inserted immediately after $\mathcal{L}\_{\text{Align}}$ is introduced, framing the frozen encoder as the anti-collapse mechanism and motivating the non-contrastive design from first principles.
> 2. **Section 4 (Theoretical Analysis):** We will formalize the proof, draw an explicit contrast with SimSiam/BYOL, and connect back to the surrogate redefinition in W1, thus closing the loop between methodology and theory.
>
> This restructuring ensures the role of the frozen encoder is established at first introduction and then rigorously formalized, rather than remaining implicit as it currently does.

---

> > ### Author Rebuttal · Reviewer_U2kh · 2026-04-03
> >
> > Most of my concerns are solved, thanks for the rebuttal

---

> > > ### Author Response · Authors · 2026-04-03
> > >
> > > Thank you very much for your thoughtful follow-up and for letting us know that most of your concerns have been resolved. We sincerely appreciate your time and consideration.
> > > We are glad that the rebuttal was helpful, and we will carefully incorporate the relevant clarifications and improvements into the final manuscript.
> > >
> > > If you feel it is appropriate, we would also sincerely appreciate your reconsideration of the score in light of the rebuttal and discussion.
> > >
> > > **Thank you again for your valuable feedback and support :)**

---

### Decision · Program_Chairs · 2026-04-30

**Decision:**

Accept (regular)

**Comment:**

Reviewers noted the interest of the experimental design of the paper and the strong empirical foundation of this contribution, in particular regarding the diversity across the datasets and baselines. They also emphasized that the paper sets the focus on an interesting and original question:  providing a link between high predictive accuracy and learnt latent representations with temporal continuity.

However, they raised several methodological concerns, in particular with respect to the VAE training, as well as the lack of guarantees that the latent space learns meaningful and structured representations.

During the rebuttal phase, the authors addressed most of these concerns. They provided many empirical experiments supporting the validity of their approach (t-SNE visualizations to support latent space structure, replacing the pre-trained autoencoder with an untrained MLP). The authors also expanded their ablation study, demonstrating in particular that the alignment loss consistently improves performance across all considered settings. They also addressed requests for quantitative metrics about the dynamical structure.

While the current derivation of the model does not explicitly enforce that the latent space captures meaningful system dynamics, the authors provided sufficient evidence that this is the case (the multi-step rollout stability results provides empirical evidence that LatentTSF learns structured and predictable dynamics).

One of the main perspectives of this work relies on incorporating explicit stability constraints or theoretical guarantees to further strengthen the contribution, the overall empirical study is convincing.